

# A Landsat-based model for retrieving total suspended solids concentration of estuaries and coasts

Chongyang Wang[1,2,3], Shuisen Chen[2], Dan Li[2], Wei Liu[1,2,3], Ji Yang[1,2,3], Danni Wang[4]

[1] Guangzhou Institute of Geochemistry, Chinese Academy of Sciences, Guangzhou 510640, China

[2] Guangzhou Institute of Geography, Guangzhou 510070, China

[3] University of Chinese Academy of Sciences, Beijing 100049, China

[4] Department of Resources and the Urban Planning, Xin Hua College of Sun Yat-Sen University, Guangzhou 510520, China

*Correspondence to*: Shuisen Chen (css@gdas.ac.cn)

**Abstract.** Retrieving total suspended solids (TSS) concentration accurately is essential for sustainable management of estuaries and coasts, which plays a key role in the interaction of hydrosphere, pedosphere and atmosphere. Although many TSS retrieval models have been published, the general inversion method that is applicable to different field conditions is still under research. In order to obtain a TSS remote sensing model that is suitable for estimating the TSS concentrations with wide range in estuaries and coasts by Landsat imageries, this study recalibrated and validated a

number of regression-techniques-based TSS retrieval models using 129 in-situ samples collected from five regions of China during the period of 2006-2013. It was found that the optimized Quadratic model using the Ratio of Logarithmic transformation of red band and near infrared band and logarithmic transformation of TSS concentration (QRLTSS) works well and shows a relatively satisfactory performance. The adjusted QRLTSS model based on Landsat sensors explain about 72% of the TSS concentration variation (TSS: 4.3-577.2 mg/L, N=84) in the study and have an

acceptable validation accuracy (TSS: 4.5-474 mg/L, RMSE: 21.5-25 mg/L, MRE: 27.2-32.5%, N=35). The QRLTSS model based on Landsat OLI is better than TM and ETM+ ($R^2$: 0.7181 vs. 0.7079, 0.708) because of the optimization of OLI sensor's design. A threshold of red band reflectance (OLI: 0.032, ETM+ and TM: 0.031) was proved capable to help solve the QRLTSS model and retrieve TSS concentration from Landsat remote sensing imageries. After 6S model-based atmospheric correction of Landsat imageries, the TSS concentrations of three regions (Moyangjiang

River Estuary, Pearl River Estuary and Hanjiang River Estuary) in Guangdong Province of China by OLI and ETM+ imageries were retrieved by the optimized QRLTSS model. As a result, the Landsat imagery inversed TSS concentrations showed good validation accuracies with the synchronous in-situ observation (TSS: 7-160 mg/L, RMSE: 11.06 mg/L, MRE: 24.1%, N=22). The TSS concentrations retrieved from Landsat imageries in the three estuaries





showed large variation (0.295-370.4 mg/L). The further validation from EO-1 Hyperion imagery showed good performance of the model (In site synchronous measurement of TSS: 106-220.7 mg/L, RMSE: 26.66 mg/L, MRE: 12.6%, N=13) for the area of high TSS concentrations in Lingding Bay of Pearl River Estuaries as well. Evidently, the QRLTSS model can be potentially applied to simulate high-dynamic TSS concentrations of other estuaries and coasts

5   in the world by Landsat imageries, improving the understanding of the spatial and temporal variation of TSS concentrations on regional and global scales. We believe that the optimized QRLTSS model can hopefully be further adjusted to establish a regional or unified TSS retrieval model of estuaries and coasts for different satellite sensors similar to Landsat OLI-ETM+-TM sensors or with similar red and near infrared bands, such as ALI, HJ-1 A/B, LISS, CBERS, ASTER, ALOS, RapidEye, Kanopus-V, GF, etc.

Keywords: total suspended solids, estuaries and coasts, QRLTSS model, evaluation and optimization, remote sensing

# 1 Introduction

Total suspended solids (TSS) is a critical factor of the ecological environment of water bodies, which directly and deeply affects their optical properties through absorbing and scattering of the sunlight (Chen et al., 2015b; Pozdnyakov

et al., 2005; Wang et al., 2016; Wu et al., 2013), leading to impacts on the primary production of the water areas (May et al., 2003). Estuaries and coasts are the most important intermediate zones that connect hydrosphere, pedosphere and atmosphere, which then pass on a deep and wide impact on many aspects of our societal and natural environment (Nechad et al., 2010; Pozdnyakov et al., 2005). The topic of TSS concentration monitoring and spatial and temporal variation assessment has been paid great attention, and the associated research work has been conducted more

frequently by a variety of scholars, government branches and society communities (Caballero et al., 2014; Giardino et al., 2015; Liu et al., 2003; Lu et al., 2012; Nechad et al., 2010; Olmanson et al., 2013; Pozdnyakov et al., 2005; Rao et al., 2009; Shen et al., 2008; Tang et al., 2004b; Zhang et al., 2007). Many methods can be used to estimate TSS concentrations of water bodies, including hydrological sites monitoring, in-situ investigation, physical models, numerical simulation, remote sensing and so on (Chen et al., 2015a). Retrieving TSS concentrations from remote

sensing data has unique advantages due to the wide spatial coverage and periodic revisit, such as Land Observation Satellite (Landsat), the first Earth Observing (EO-1), the Moderate Resolution Imaging Spectroradiometer (MODIS), the Medium Resolution Imaging Spectrometer (MERIS), the Geostationary Ocean Color Imager (GOCI), the Sea



Viewing Wide Field of View Sensor (SeaWiFS), Systeme Probatoire d'Observation dela Tarre (SPOT) and the Environment and Disaster Monitoring and Forecasting Small Satellite Constellation (HJ). Compared with other remote sensing data, Landsat series of imageries with additional good quality, high spatial resolution and inheritance, especially long-term historical data since 1972, has advantage on spatiotemporal dynamics analysis of TSS

concentrations (Wu et al., 2013).

Many Landsat-based models have estimated the TSS concentration with empirical, semi-empirical, semi-analytical or analytical algorithms (Chen et al., 2014; Doxaran et al., 2003; Fraser., 1998; Islam et al., 2001; Li et al., 2010; Nas et al., 2010; Oyama et al., 2009; Raharimahefa and Kusky., 2010; Rao et al., 2009; Ritchie and Cooper., 1991; Topliss et

al., 1990; Volpe et al., 2011; Wang et al., 2016; Wu et al., 2013; Zhang et al., 2014). Based on the rigorous theoretical derivation, the semi-analytical and analytical models are likely more applicable to different water bodies than the empirical or semi-empirical methods (Binding et al., 2012; Binding et al., 2010; Chen et al., 2015b; Giardino et al., 2007; Ma et al., 2010; Sipelgas et al., 2009). However, there are still limitations of the application due to the difficulties of retrieval or inaccuracies on initialization parameters (Binding et al., 2012; Chen et al., 2015b; Ma et al.,

2010; Wu et al., 2013). Therefore, the empirical, especially semi-empirical methods are still used to estimate TSS concentration due to easy-to-use, and will continue to be used for a long time with advantages of simplicity and sufficient accuracies. It should be noted that the applications of empirical or semi-empirical TSS models need to be revalidated in different regions and periods because they are largely region-, time- or environment dependent (Wu et al., 2013). Thus, we could find that previous empirical or semi-empirical Landsat-based TSS retrieval models vary greatly

in the forms of models (Ma et al., 2010; Wu et al., 2013).

The TSS models of single band include linear function (Fraser., 1998; Islam et al., 2001; Nas et al., 2010; Rao et al., 2009), exponential or logarithmic function (Keiner and Yan., 1998; Wu et al., 2013; Zhang et al., 2014) and quadratic function (Chen et al., 2014). Those models have been applied to many regions easily because they have not only the

simple forms but more choice of remote sensing data. However, we all know that the sensitivity of satellite sensor bands is different for different TSS concentrations. Many studies have proven that reflectance in the red band increases with increasing of TSS concentrations but is apt to convergence or keeps stable due to saturation effect of high TSS concentrations (Ritchie and Zimba., 2005; Feng et al., 2014), while the reflectance in the near infrared band is sensitive to high TSS concentrations although not sensitive for low TSS concentrations (Chen et al., 2015b; Feng et al., 2014;



Hu et al., 2004; Wang et al., 2010). Thus, those models of single band have limited of applications in regions with wide dynamic TSS concentration range.

Models of multiple bands combination work better than single band in avoiding the effect of saturation for water
bodies of high TSS concentrations. We can find that the models of multiple bands combination were applied to estimate TSS concentration frequently (Dekkera et al., 2001; Doxaran et al., 2003; Feng et al., 2014; Oyama et al., 2009; Wang et al., 2016). Although the combination includes band ratio (Doxaran et al., 2003; Lathrop et al., 1991; Ritchie and Cooper., 1991; Topliss et al., 1990; Wang et al., 2016) and many other complex combination of multiple bands (Dekkera et al., 2001; Li et al., 2010; Oyama et al., 2009; Song et al., 2011; Zhang et al., 2015), the models of
multiple bands combination can be also classified into linear, exponential, logarithmic and quadratic functions. It should be noted that most of those empirical or semi-empirical TSS retrieval models are simple monotonic functions except quadratic forms of models (Chen et al., 2014; Ritchie and Cooper., 1991; Topliss et al., 1990). One potential issue of monotonic function is that a little change of band reflectance can cause exaggerated estimation of TSS concentrations. Although some non-monotonic functions could avoid the exaggerated issue effectively, it is widely
believed that there does not exist a regional or universal empirical or semi- empirical TSS retrieval model for all water bodies currently (Ma et al., 2009; Tang et al., 2005; Wu et al., 2013).

Recently, we published a conference paper (Wang et al., 2016), which stated that the quadratic TSS model seems to be a new non-monotonic function for estimating TSS concentrations of multiple estuaries and coasts. However, the model
lacks necessary analysis of use condition and discussion of results due to limitation of space. What's more, the model in our previous work (Wang et al., 2016) could not be applied to remote sensing data due to one major drawback: each value of reflectance does not correspond to a unique TSS concentration based on the model. And, the accuracy of model was not validated from Landsat imageries or other remote sensing sensors, which is critical to the robustness and further application of TSS retrieval model in practice.

Based the above analysis, this study intends to further evaluate and validate the Landsat-based model for retrieving TSS concentrations in estuaries and coasts, and improve our previous work (Wang et al., 2016). In order to achieve this objective, the applicability of more than 20 previous Landsat-based models was reviewed and further analyzed. And we focus on the models examination of multiple bands combination, which belong to non-monotonic function. The



issue of the model (Wang et al., 2016) that could not be applied to remote sensing data was solved by providing the TSS concentration division of vertex value of quatratic function. This paper was organized as follows. In-situ data and Landsat imageries were described along with the atmospheric correction method and assessment method of simulation model accuracy in Section 2. The TSS retrieval model, validation and the spatial analysis of TSS concentration mapped from Landsat imageries and EO-1 Hyperion imagery were presented in Section 3. Finally, the summary and conclusions were given in Section 4.

# 2 Materials and study methods

## 2.1 Study areas

The study areas include five regions of China listed as follows.

Region a, Xuwen coast (Fig.1a), located between longitudes 109.8°~110.1°E and latitudes 20.1°~20.5°N, is the important Coral Reefs National Nature Reserve with the most plentiful coral species because of its less turbid waters. The good quality waters are due to less water discharge ($2.74 m^3$/s) and suspended sediment load ($3*10^4$ tons with annual mean) (Chen et al., 2015b; Wang et al., 2002). However, it was reported that the coral reefs does not grow as well as before (Zhao et al., 2011). Researchers believe that it is mainly caused by the increasing TSS concentration, declination of water transparency and decreasing water temperature due to excessive fish farming, overfishing and industrial pollution (Chen et al., 2015b). Besides, the coastal land development is an important reason.

Region b, Moyangjiang River Estuary (Fig.1b), is located between longitudes 112°~112.2°E and latitudes 21.65°~21.9°N, southwest of Guangdong Province. The source of Moyangjiang River in Yangchun County, and it has a length of 199 kilometers and a drainage area of more than 6000 square kilometer. Moyangjiang River crosses Yangchun, Yangdong and Jiangcheng Counties (Districts) and flows into the South China Sea.

Region c, Pearl River Estuary (Fig.1c), is located between longitudes 113.15°~114.1°E and latitudes 21.9°~23°N. Pearl River has the fourth longest and largest drainage area in China and its annual runoff is smaller than Yangtze River only. It crosses eight water ways (Humen, Jiaomen, Hongqimen, Hengmen, Modaomen, Jitimen, Hutiaomen and Yamen)



located at six cities of Guangdong Province and pours into South China Sea. As we all know estuary and coast of Pearl River suffers from severely combined pollution (Ma and Wang 2003) which mainly comes from industrial production, residential life and seawater intrusion (Chen et al., 2009a).

Region d, Hanjiang River Estuary (Fig.1d), is located between longitudes 116.6°~117°E and latitudes 23.2°~23.6°N, east of Guangdong Province and southwest of Fujian Province. Hanjiang River has the second largest drainage area in Guangdong Province. The lower reaches of Hanjiang River include Beixi water way located in northeast, Dongxi water way located in the middle and Xixi water way located in west. Xixi water way also crosses with the three water ways of Waishahe, Xinjinhe and Meixi and flows into the South China Sea. Waishahe, Xinjinhe and Meixi water ways
are located in east, middle and west of Longhu District, Shantou of Gungdong Province, respectively.

Region e, Yangtze River Estuary (Fig.1e), is located between longitudes 121.55°~122.4°E and latitudes 30.8°~31.8°N. Yangtze River is the largest river in China. The annual mean surface runoff is $9.2*10^{11}m^3$ and suspended sediment load is about $4.8*10^8$ tons (Feng et al., 2014). Such huge terrestrial input not only loads to its extremely turbid waters but
also impacts on the optical properties of this region. It is reported that the environment of Yangtze River estuary is getting worse due to the rapid developments and urbanization in the surrounding industrial areas (Chen et al., 2015a; Hsu and Lin., 2010). As a result, there are more and more studies focusing on this region due to its important ecological and economic role (Chen et al., 2015a; Feng et al., 2014; Shen et al., 2010).

## 2.2 In-situ data and satellite data

The 129 in-situ samples were collected from the above-mentioned five regions of China, whose positions were recorded by Trimble global positioning system with root mean square errors of about 1~4 m (shown in Fig.1). We used these samples to establish the spectral model of TSS retrieval for estuaries and coasts. It includes 32 samples from Xuwen coast on December 3, 2010 and February 13~14, 2013, 11 samples from Moyangjiang River Estuary on December 6, 2013, 40 samples from Pearl River Estuary on December 19, 21, 2006, December 27, 2007 and
November 2, 2012, 12 samples from Hanjiang River Estuary on December 1, 2013 and 34 samples from Yangtze River Estuary on October 14~15, 2009. The field spectral measurements and synchronous water samples of all 119 sites were carried out from 10:00 to 15:00 (Fig.1, dots. Table 1). Another ten samples with TSS only from Pearl River Estuary on



December 21, 2006 were collected synchronous with Hyperion imagery (Fig.1c, triangles. Table 1). The reflectance were measured based on above-water spectrum measurement method (Tang et al., 2004a) which was applied to the water bodies like estuaries and coasts of China widely. Finally, the reflectance of water surface (Fig.2) was calculated in the same way as Zhang et al (2014) and Chen et al (2015b) did while TSS concentration was measured from water

samples by a weighed method (Binding et al., 2012; Caballero et al., 2014).

Sensors of TM, ETM+ and OLI aboard the satellite Landsat 5, 7 and 8 have a spatial resolution of about 30m, with more than seven band spanning from visible to infrared wavelength. Landsat imagery can be fit for quantifying the optical properties in oceans, lakes, estuaries and coasts which have been explored in many studies although they were

originally designed for observation of land targets (Zhang et al., 2014).

This study only obtained three Landsat imageries with good quality, which can be matched with synchronous in-situ measurements of three regions (Table 1) of the study areas due to frequent cloud coverage in estuary and coast of Rivers and the low temporal resolution (16d) of TM, ETM+ and OLI (Bailey and Werdell., 2006). The first scene of

image from ETM+ (path/row = 122/45) was captured on November 2, 2012, covering part of Pearl River Estuary (Fig.1c). The second scene of image (path/row = 120/44) from OLI was captured on December 1, 2013, covering Hanjiang River Estuary (Fig.1d). The third scene of image (path/row = 123/45) from OLI was captured on December 6, 2013, covering Moyangjiang River Estuary (Fig.1b). We noted that the scan line corrector (SLC) of Landsat 7 ETM+ has failed since May 31, 2003. However, there are still many research works using the SLC-off data which is repaired

using means of methods such as local self-adaptive regression analysis (Zhang et al., 2014). The repaired data in our study is provided by the International Scientific Data service Platform, Computer Network Information Center, Chinese Academy of Sciences (http://datamirror.csdb.cn).

In addition, a scene of Earth-observing (EO-1) Hyperion image (path/row=122/44) was captured on December 21,

2006 with 13 synchronous in-situ samples, covering part of Pearl River Estuary (Fig.1c). With spectral coverage ranging from 400 to 2500 nm and 10 nm (sampling interval) of contiguous bands of the solar reflected spectrum, Hyperion's spatial resolution is 30 m with a 7.7 km imagery swath and 185 km length (http://eo1.usgs.gov). Hyperion is also well suited for retrieving spatial distributions of water-color constituents in Pearl River Estuary (Chen et al., 2009a). The Hyperion data was just used for further and supplementary validation of TSS retrieving model here.



## 2.3 Atmospheric correction method

Atmospheric correction is a necessary step before the application of remote sensing imageries (Gordon and Wang., 1994). The commonly used methods of atmospheric correction include the simple Dark Object Subtraction (DOS), Fast Line-of-sight Atmospheric Analysis of Spectral Hypercubes (FLAASH), and Second Simulation of Satellite

Signal in the Solar Spectrum (6S) models. In addition, the regression analysis is also a common method which is based on the correlation among spectral bands of sensors (Chen et al., 2011a; Mei et al., 2001).

The 6S code is the improved version of Simulation the Satellite Signal in the Solar Spectrum (5S), developed by the Laboratoire d'Optique Atmospherique (Vermote et al., 1997). It was frequently used for atmospheric correction of

Landsat sensors based on the Landsat Ecosystem Disturbance Adaptive Processing System (LEDAPS, a Landsat atmospheric correction codebase funded by NASA's Terrestrial Ecology Program) (Feng et al., 2013; Ju et al., 2012; Maiersperger et al., 2013; Masek et al., 2006). Thus, the LEDAPS software was chosen for atmospheric correction in this paper. We assumed the continental aerosol type because the northeast monsoon was blowing from the land in the study. The aerosol optical thickness was derived independently from each Landsat acquisition using the dark dense

vegetation (DDV) approach (Kaufman and Tanré, 1996). Critical atmospheric parameters of 6S model, including water vapor at a resolution of 2.5 by 2.5 degrees (http://dss.ucar.edu/datasets/ds090.0/) and ozone concentrations at a resolution of 1.25 longitude and 1 latitude, were collected from National Centers for Environmental Prediction (NCEP) and Total Ozone Mapping Spectrometer (TOMS) or NOAA's Television Infrared Observation Satellite Program (TIROS) Operational Vertical Sounder (TOVS), respectively (Feng et al., 2013; Ju et al., 2012; Masek et al., 2006).

Rayleigh scattering was adjusted to local conditions by a static 0.05 degree digital topography dataset (derived from the 1 km GTopo30) and NCEP surface pressure data (Feng et al., 2013; Masek et al., 2006). All the parameters are automatically called corresponding to each Landsat image when LEDAPS runs. The global surface reflectance products from Landsat of high quality had been obtained using LEDAPS implementation of 6S model (Feng et al., 2013; Maiersperger et al., 2013).

## 2.4 Assessment method of accuracy

In order to validate the accuracy of the TSS models, atmospheric correction of remote sensing imageries and mapping



of TSS from remote sensing imageries, the determination coefficient ($R^2$), the root mean square error (RMSE) and mean relative error (MRE) of modeled to measured values are used to assess the accuracy. RMSE and MRE use the following equations, respectively.

$$RMSE = \sqrt{\frac{\sum_{i=1}^{i=n}\left(x_i - x_i'\right)^2}{n}} \qquad (1)$$

$$MRE = \frac{\sum_{i=1}^{i=n}\left|\frac{x_i - x_i'}{x_i}\right|}{n} \times 100\% \qquad (2)$$

Where $x_i$ is the observed value, $x_i'$ is the modeled value, $i$ is the i th element, and $n$ is the number of elements.

# 3 Results and discussion

## 3.1 Review of previous Landsat-based TSS retrieval models

Before establishing TSS retrieval model, the water surface reflectance measured in the field was convoluted with the Landsat band response functions to derive the band-weighted reflectance data using equation (3). It is also a critical step for the application of TSS retrieval model from ground spectral data to remote sensing imageries.

$$R\left(\text{band}\right) = \frac{\sum_{band_{min}}^{band_{max}} f(\lambda_{band}) r(\lambda_{band})}{\sum_{band_{min}}^{band_{max}} f(\lambda_{band})} \qquad (3)$$

Where $band_{min}$ and $band_{max}$ are the lower and upper limits of Landsat band in red and near infrared bands, respectively. The $f(\lambda)$ is the spectral response function of Landsat sensors (http://Landsat.usgs.gov). Thus, two simulated "$R$(band)", corresponding to the two Landsat bands (red band and near infrared band), were calculated for each samples.

Firstly, the previous Landsat-based TSS retrieval models (Table 2) were calibrated and validated again with the adjustment of parameters based on the 119 in situ samples (84 in situ samples for calibration, the other 35 for





validation), shown in Fig.3. The results indicated that the previous Landsat-based TSS models do not explain the TSS variation so well when they were used to estuaries and coasts in the study. The determination coefficient of calibration models in the five best TSS models (Fig.3) is between 0.58 and 0.784, corresponding to linear (Fig.3c1) and quadratic (Fig.3a1) model of single band, respectively.

Based on the other 35 in situ samples with range of 4.5-474 mg/L, we further validated the five TSS models. The results showed that minimum MRE is 39.4% from exponential model of single band (Fig.3d2) but its RMSE is 50.26 mg/L. While the quadratic model of single band (Fig.3a2) got minimum RMSE (35.73 mg/L) with high MRE value of 144.2%. Obviously, the quadratic model of single band is hard to be adopted for this study. However, to exponential

model of single band, its high RMSE prevents this form of model from its application as well, especially when we take the TSS concentration of 22 validation data lower than 36 mg/L (Fig.3 triangles) into account. The RMSEs and MREs of the other three forms of models are 69.3 mg/L and 45%, 82.7 mg/L and 48% for linear model (Fig.3b2 and c2), 68.7 mg/L and 41.3% for quadratic model (Fig.3e2), respectively. In contrast, the non-monotonic function, quadratic model of ratio of bands (Fig.3e) has a better performance among the five previous TSS models. We still expect that there

would be a TSS model with high calibration and validation accuracy for estuaries and coasts of China by Landsat imageries simultaneously.

## 3.2 Development of QRLTSS model

In order to develop a Landsat-based model with higher calibration and validation accuracy, some MODIS-based TSS retrieval models (Chen et al., 2009b; Chen et al., 2011ab; Chen et al., 2015b; Wang et al., 2010) were referred. Those

models made full use of the relationship between the ratio of logarithmic transformation of red band and near infrared band and logarithmic transformation of TSS concentration. Thus, following the feature of those MODIS based model we further optimized the model (Fig.3e1) developed by Ritchie and Cooper. (1991). The red band and near infrared band were processed with logarithmic transformation. Then, the previous model (Fig.3e1, Ritchie and Cooper (1991)) was evaluated and adjusted for this study based on relationship between the ratio of logarithmic transformation of red

band and near infrared band and logarithmic transformation of TSS concentration under Matlab environment, shown in Fig.4.

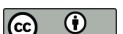



From Fig.3 and Fig.4 (a1, b1 and c1), we found that the quadratic model of the ratio of logarithmic transformation of red band and near infrared band and logarithmic transformation of TSS concentration (QRLTSS, equation (4)) has a higher calibration accuracy than most of the previous TSS models, no matter whether it is Landsat OLI, ETM+ or TM sensor.

$$\frac{Log(R_1)}{Log(R_2)} = a*(Log(TSS))^2 + b*Log(TSS) + c \qquad (4)$$

Where, $R_1$ and $R_2$ represent near infrared band and red band of OLI, ETM+ and TM sensors. Parameters $a$, $b$ and $c$ refer to Fig.4, respectively. The unit of TSS concentration is in mg/L.

Compared to the previous model developed by Ritchie and Cooper (1991), we improved the input with logarithmic transformation of bands and made full use of the different sensitivity of red and near infrared bands (Part 1, Introduction) to TSS concentrations that have been proved by many studies (Chen et al., 2015b; Feng et al., 2014; Wang et al., 2010; Hu et al., 2004). Compared with the MODIS-based models developed by Chen et al (2011b) and Wang et al (2010), the QRLTSS model established in this paper is seemingly more complex. The models (Chen et al., 2011b; Wang et al., 2010) are in linear or exponential form, belonging to simple monotonic function that can cause exaggerated estimation in some spectral range. Although the quadratic model developed in our study is similar to previous studies (Chen et al., 2009b; Chen et al., 2011a; Chen et al., 2015b), there are some differences among them. They are all quadratic models, but the models developed by Chen et al (2009b) and Chen et al (2011a) are part of the curve, different from the form developed by Chen et al (2015b) and this study, which are all complete quadratic curve. However, the part of quadratic model has limitation in estimating TSS concentration with the wide range. Some regions with lower or higher TSS concentration could not be retrieved accurately. In fact, TSS concentration developed in Apalachicola Bay, USA, the study area of Chen et al (2009b) and Chen et al (2011a), is not as high as TSS concentrations in Yangtze River Estuary, part of our study areas and the study areas by Chen et al (2015b). The maximum TSS concentration in the previous studies was about 200 mg/L (Chen et al., 2009b; Chen et al., 2011a), but The maximum TSS concentration was more than 500 mg/L for previous study (Chen et al., 2015b) and the study. In addition, the study areas of Chen et al (2015b) only include Xuwen Coral Reef National Nature Reserve, a less turbid region, and Yangtze River Estuary, an extremely turbid region, which might make the model developed by Chen et al (2015b) work worse in the middle of the quadratic curve than both ends of the quadratic curve. The QRLTSS model in



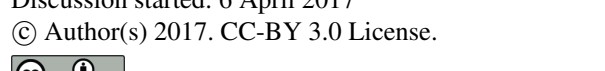

the study is better in the continuity of calibration and validation data than Chen et al (2015b). The reason is that the study areas include not only their regions (Chen et al., 2015b) but also other three main estuarine regions, Moyangjiang River Estuary, Pearl River Estuary and Hanjiang River Estuary, Guangdong province. In general, the TSS concentrations in the three additional regions are higher than Xuwen coast, but lower than Yangtze River Estuary.

This data is good supplement for the robustness of an accurate model on the previous study.

It should be noted that a band value corresponds to two TSS concentrations values based on the quadratic model (equation (4)). We should make the unique choice when validating or retrieving TSS concentration by the model from remote sensing imageries. Unfortunately, we had not been aware of this problem and did not solve it in our previous

work (Wang et al., 2016). It meant that our previous results were not complete. In this study, the TSS concentrations of vertex corresponding to the three quadratic models based on Landsat OLI, ETM+ and TM sensor have been obtained through solving the equation (Fig.4a1, b1 and c1). The vertex values are about 36.1 mg/L for OLI and about 32.2mg/L for ETM+ and TM, which attribute to their almost same spectral band features (Table 4). For QRLTSS model of OLI sensor, we found that the values of OLI red band weighted reflectance of all validation data are lower than 0.032 when

their TSS concentration is less than 36.1 mg/L (Fig.5a, blue dots) except for one exceptional data (Fig.5a, black dot). And the reflectance is higher than 0.032 when TSS concentration is more than 36.1 mg/L (Fig.5a, red dots). The QRLTSS models of ETM+ and TM sensor have similar situations. But the values of TSS concentration and reflectance at vertex are 32.2 mg/L and 0.031, shown in Fig.5b (dots for ETM+ and triangles for TM). The findings are different from the result found in MODIS-based model by Chen et al (2015b). In previous study (Chen et al., 2015b), the values

of TSS concentration and reflectance of MODIS red band at vertex are 31 mg/L and 0.025, respectively. We believe that the difference was caused by different spectral characteristics of satellite sensors (Table 4). This is also why multi-source satellite remote sensing becomes more and more important in recent years. Monitoring TSS concentration from multiple data sources could make full use of the advantages of all kinds of satellite sensors. According to the above analysis, TSS concentration can be retrieved by using equation (5) in the form of positive squared root if the

reflectance of red band is lower than 0.032 (OLI sensor) or 0.031 (ETM+ and TM sensors), and using equation (5) in the form of negative squared root if the reflectance of red band is greater than 0.032 (OLI sensor) or 0.031 (ETM+ and TM sensors), respectively.



$$Log(TSS) = \frac{-b \pm \sqrt{b^2 - 4a\left(c - \frac{Log(R_1)}{Log(R_2)}\right)}}{2a}, \left(b^2 - 4a\left(c - \frac{Log(R_1)}{Log(R_2)}\right) \geq 0\right) \quad (5)$$

We validated the QRLTSS model based on the 35 in situ samples and the selection criteria mentioned above, equation (5). The results indicated that the QRLTSS model has a better performance than the previous five TSS models although QRLTSS model explained no more than 72% of the TSS concentration variation. The RMSEs and MREs of all validation data for QRLTSS model are 21.5 mg/L and 27.2% for OLI (Fig.4a2), 25 mg/L and 32.5% for ETM+ (Fig.4b2), and 24.9 mg/L and 31.5% for TM (Fig.4c2), respectively. All of the simulated results from QRLTSS model have higher validation accuracies than the best (RMSE: 35.7 mg/L, MRE: 39.4%) of the five previous TSS models. In order to get better validation result, the wide range of validation data (TSS: 4.5-474 mg/L) was divided into two parts of low (4.5-32.2 mg/L, triangles in Fig.3 and Fig.4) and high (36.2-474 mg/L, squares in Fig.3 and Fig.4) TSS concentration for further validation according to the vertex location of the quadratic model. For the data of low TSS concentrations, the RMSEs and MREs of validation are 3.5 mg/L and 31.1% for OLI, 4.6 mg/L and 38.3% for ETM+, and 4 mg/L and 35.3% for TM. For the data of high TSS concentrations, the RMSEs and MREs of validation are 40.6 mg/L and 25.1% for TM, 40.7 mg/L and 20.3% for ETM+, and 35.1 mg/L and 20.7% for OLI. The validation accuracies of the two parts are still better than the best (RMSEs & MREs: 5.6 mg/L & 39.4% for low concentration part, and 53.5 mg/L & 23.5% for high concentration part) of the previous five TSS models. The detailed information of calibration and validation in Fig.3 and Fig.4 were shown in Table 3.

From Table 3 we could also find that the calibration and validation accuracy of OLI-based QRLTSS model is a little better than ETM+ and TM ($R^2$: 0.7181 vs 0.708 and 0.7079). We attribute this mainly to the improvement of Landsat OLI sensor's design. Especially in OLI band_5 used in our TSS retrieval model, the band of water vapor absorption at 825 nm were removed from the near infrared band range, whose wavelength is 845-885 nm (http://Landsat.usgs.gov/Landsat8.php). And the near infrared band wavelength of ETM+ is 775-900 nm, and 760-900 nm for TM. In addition, the red band wavelength of OLI is 630-680 nm, and 630-690 nm for ETM+ and TM. The little difference of sensors determines a little difference in QRLTSS model of TSS for ETM+ and TM. The performance (red band and near infrared band) and the vertex of quadratic model based on different sensors were shown in Table 4.



## 3.3 Comparison of Landsat measured reflectance with in-situ reflectance

In order to analyze the spatial and temporal variation of TSS concentrations in our study areas and further verify accuracy of QRLTSS model, the acquired Landsat imageries were used to calculate the TSS concentrations by this model described in Section 3.1. (Equation 5). First of all, Landsat images were well calibrated by atmospheric correction of 6S, which is critical for working with multi-scene imageries and using empirical/semi-empirical method. Then the atmospheric correction accuracy of 6S was calculated based on the reflectance of synchronous in situ measurements, a total of 22 samples from three regions within two-hour time window of satellite overpass. Six of the total 22 samples from Pearl River Estuary were from November 2, 2012, nine samples from Hanjiang River Estuary were from December 1, 2013 and the other seven samples from Moyangjiang River Estuary were from December 6, 2013. In deriving the reflectance comparison, the water-leaving radiances from Landsat imageries were averaged by windows of 3x3 pixels when they were compared with in situ measurement. We then calculated RMSE and MRE of the reflectance result of atmospheric correction with in situ reflectance. RMSEs and MREs of red and near infrared bands are 0.0033, 9.58% and 0.00092, 21.5%, respectively, which showed an acceptable accuracy. Fig.6 shows that the 6S model is sufficiently stable and accurate for deriving the reflectance at visible and near infrared band from broadband satellite data for the purpose of remote sensing applications in estuarine and coastal waters.

## 3.4 Analysis and validation of QRLTSS mapping from Landsat imageries

After atmospheric correction of 6S, the TSS concentration of Moyangjiang River Estuary, part of Pearl River Estuary, Hanjiang River Estuary were estimated from ETM+ or OLI imagery (Fig.7). Water bodies were extracted based on method developed by Jiang at el (2014). The difference of spectral profile across water and cloud were used to mask clouds (Chen et al., 2009; Chen et al., 2011a). In estuary and coast of Pearl River, we find that the reflectance of water is usually less than 0.05 in near infrared band while the reflectance of cloud is usually more than 0.1 in near infrared band. Thus, the clouds were masked based on reflectance that is more than 0.05 in near infrared band.

In Fig.7a, the TSS concentration in Moyangjiang River Estuary (Beijing time at 11:00), on December 6, 2013 showed a large variation ranging from 0.557 mg/L to 203.9 mg/L. It is very clear that the TSS concentrations are higher inside and outside of Moyangjiang River Estuary than outer shelf area, especially in the estuary downstream, with a mean



value of 154.2 mg/L (Fig.7a). The region of high TSS concentrations in Moyangjiang River Estuary looks lung-shaped. And the outer shelf area has low TSS concentrations, where the TSS concentrations less than 35 mg/L were frequently found and the maximum is not more than 60 mg/L. So, there is a sharp front that could be seen clearly between coastal area and outer shelf area. The TSS distribution in Moyangjiang River Estuary could attribute to interaction of tide and

runoff. Remote sensing imagery covering Moyangjiang River Estuary was obtained at 11:00 in the morning when the tide had begun to ebb and runoff with large amounts of sediment flowing into the South China Sea.

Different from TSS concentrations in Moyangjiang River Estuary, the TSS concentrations in eastern Zhuhai & Macao and Hongkong coastal water bodies are much lower with a mean value of 12 mg/L (Fig.7b, Blank areas without

synchronous image). There is a significant decrement trend of TSS concentration from the northwest to southeast of Pearl River Estuary. It is mainly due to the interaction between runoff (flowing towards southwest) and tide (flowing towards northwest). The maximum TSS concentration is about 29 mg/L. The reason why the water bodies in outer Lingding Bay of Pearl River Estuary have a low level TSS concentration is mainly because of strong management protection and less human activity. Most part of eastern Zhuhai water bodies belong to Pearl River Estuary Chinese

White Dolphin National Nature Reserve since 2003, which is about 460 square kilometers located between longitudes 113.66°~113.87°E and latitudes 21.18°~22.4°N, shown in Fig.7b (Region with black dotted line, http://www.gdofa.gov.cn/). The low TSS concentrations in this region confirm the protection effect of Chinese White Dolphin.

Compared to Moyangjiang River Estuary and eastern Zhuhai and Macao coastal water bodies, the TSS concentrations of Hanjiang River Estuary had wider variables, ranging from 0.295 mg/L to 370.4 mg/L. However, the water bodies with high TSS concentrations in this region mainly were in two zones where the sharp front is clearly visible (Fig.7c, 1 and 2). The TSS concentrations in zone 1 are almost more than 100 mg/L with maximum value of 370.4 mg/L and a mean value of 167.91 mg/L. For zone 2, the TSS concentration mainly ranged from 20 mg/L to 110 mg/L, and the

maximum and mean value are 127.14 mg/L and 61.57 mg/L. The results also showed that the turbid river runoff flows into South China Sea along east coast of Dahao District, Shantou City. The high TSS concentrations in this region were caused by different factors. In zone 1 at opposite bank of Dahao District, Shantou, it was mainly caused by the runoff of Xixi waterway, Hanjiang River and flow guiding line (Solid black line in Fig.1d and Fig.7c) connected to Longhu District, Shantou City. While in zone 2, the high TSS concentrations resulted from the interaction of tide current and



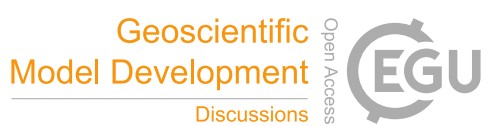

runoff, where is the potential location of estuarine barrier bar. The TSS concentrations in estuary of Xinjinhe waterway, Hanjiang River are less than 50 mg/L. Our finding is similar to the result of Ding and Xu (2007), which showed TSS concentrations ranged from 0.1 mg/L to 300 mg/L in Hanjiang River Estuary.

5      We also assessed the results of TSS concentration estimated from Landsat imageries (two OLIs and one ETM+) with in situ measurement in Pearl River Estuary, Hanjiang River Estuary and Moyangjiang River Estuary. The validation accuracy was shown in Fig.7d. RMSE and MRE of comparison of 22 field TSS concentration (7-160 mg/L) between Landsat satellite inversion and in-situ measurements of three estuaries are 11.06 mg/L and 24.1%, which is applicable to the mapping of TSS in the three Estuaries.

10    ## 3.5 Further validation of QRLTSS model from EO-1 Hyperion imagery

We were fortunate to acquire a scene of EO-1 Hyperion imagery at 10:33 (Beijing time) with synchronous TSS samples (N=13) on December 21, 2006 covering part of Pearl River Estuary from northeast to southwest of the Lingding Bay. The 13 samples (ten sites with symbol of triangles and three sites with symbol of dots, Fig.1c) were collected in footprint of Hyperion image within two-hour time window of EO-1 satellite overpass. The data set gives

15    us an opportunity to further validate the accuracy of the QRLTSS model although there are some differences between the inverted results of Hyperion and Landsat imagery. After similar pre-processing steps with Landsat imageries (Part 2.3), the 31st band (660.85 nm) and the 48th band (833.83 nm) of the Hyperion imagery corresponding to red and near infrared band of Landsat ETM+ were selected to retrieve TSS concentrations using the QRLTSS model. The results of TSS concentration mapping and validation accuracy were shown in Fig.8.

The TSS concentration mapping from Hyperion image on December 21, 2006 ranged from 1.79 mg/L to 361.6 mg/L, with a mean value of 124.4 mg/L (Fig.8a). The mapping results of TSS showed large variation from northeast to southwest in Pearl River Estuary. The areas of low TSS concentration were detected near the west of Lingding Bay (mostly in Pearl River Estuary Chinese White Dolphin National Nature Reserve, Fig.7b) and in deep channels (East

25    Channel and West Channel, Fig.8a) of Lingding Bay. The areas of high TSS concentration were in accord with the outlets of different waterways (Humen, Jiaomen, Hongqimen and Hengmen) of Pearl River Estuary frequently or the foreshores, which indicate that the maximum turbidity zones of the estuary. The 13 synchronous samples of TSS were



mostly collected from the northern zone of high TSS concentrations ranging from 106 mg/L to 220.7 mg/L (Fig.1c). Comparisons of accuracy validation between in situ and Hyperion imagery inversed TSS concentrations were produced in Fig.8b. The RMSE and MRE of comparison are 26.66 mg/L and 12.6%, respectively. It showed that the QRLTSS model also worked well in area of high TSS concentrations from Hyperion mapping result of Pearl River Estuary.

So, we conclude that QRLTSS model has the advantage for quantitative inversion of TSS concentrations with a high dynamic range in estuaries and coasts, based on the results of calibration and validation of TSS spectral model (Fig.4) and the TSS OLI, ETM+ and Hyperion remote sensing mapping (Fig.7 and Fig.8). These results explained the usability of QRLTSS model by the validation of multi-spectral Landsat OLI/ETM+ and hyperspectral EO-1 Hyperion imageries compared to our previous work (Wang et al., 2016).

# 4 Summary and conclusions

This study evaluated and optimized a quadratic model using the ratio of logarithmic transformation of red band and near infrared band and logarithmic transformation of TSS concentration (QRLTSS) for estimating TSS concentration of estuaries and coasts from Landsat imageries. The QRLTSS spectral model had reasonable accuracy for wide TSS concentration variables in estuaries and coasts (4.3-577.2 mg/L, N=84; $R^2$: ~0.72). In addition, the optimized QRLTSS model was further validated by the independent in-situ samples, and we found that the optimized model had the best validation accuracy among the 25 examined TSS retrieval models (Table 2 and 3), for the whole (4.5-474 mg/L, N=35), low (4.5-32.2 mg/L, N=22) and high (36.2-474 mg/L, N=13) range of TSS concentrations. It was found that the QRLTSS spectral model based on the bands of OLI showed a higher accuracy than bands of ETM+ and TM (Table 3 and Fig.4), which can be explained by the adjusted band design of OLI sensor in reducing the effect of water vapor absorption compared to ETM+ and TM sensors (Table 4).

It was also found that the QRLTSS spectral model showed good performance when it was applied to estimate TSS concentrations from Landsat OLI/ETM+ imageries (Fig.7). The RMSEs and MREs of validation accuracy in Moyangjiang River Estuary (December 6, 2013), part of Pearl River Estuary (November 2, 2012) and Hanjiang River Estuary (December 1, 2013) are 11.06 mg/L and 24.1% for the whole range (7-160 mg/L, N=22) of TSS

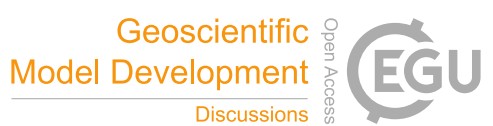

concentrations, 3.75 mg/L and 22% for the low range (7-28.2 mg/L, N=18), and 24.69 mg/L and 33.2% for the high range (37-160 mg/L, N=4) of TSS concentration, respectively. In addition, the high validation accuracy of TSS mapping from Hyperion imagery of Pearl River estuary (December 21, 2006) with in situ data (106~220.4 mg/L) using the QRLTSS model had also been obtained (Fig.8, RMSE: 26.66 mg/L, MRE: 12.6%, N=13).

Landsat imageries could be one of the best choices in terms of the availability of data source for TSS remote sensing in estuaries and coasts, considering the spatial resolution of ~30m and acquirement of long time series since 1972. The research shows that the optimized QRLTSS model can quantify the TSS concentration variation of estuaries and coasts by Landsat series of imageries with applicable accuracies ($R^2$: 0.71-0.72, 30m), which can be compared to the accuracies of previous OLI/ETM+/TM based studies ($R^2$: 0.67-0.92, 30m (Chen et al., 2014; Nas et al., 2010).) and MODIS-based studies ($R^2$: 0.61-0.86, 250m, (Chen et al., 2011a; Wang et al.,2010)). Besides, we found that the TSS concentrations (Fig.5, OLI: 36.1 mg/L, ETM+ and TM: 32.2 mg/L) at vertex of QRLTSS model based on Landsat sensors are different from MODIS (31 mg/L). Based on the vertex of QRLTSS model, we proposed a threshold (corresponding to the vertex of quadratic function) of red band reflectance (Fig.5, OLI: 0.032, ETM+ and TM: 0.031) which can be used to divide the quadratic function for solving the improved QRLTSS model under two kinds of squared roots (Table 4).

For a lot of medium- and high- resolution remote sensing sensors similar to Landsat series satellites, such as HJ-1 A/B, LISS, CBERS, ASTER, ALOS, RapidEye, Kanopus-V, GF, etc, we deduce that there is potential to adjust the QRLTSS model for mapping the wide range TSS concentrations of estuaries and coasts. Therefore, it will be beneficial to remote sensing communities if the QRLTSS model can be further calibrated and validated based on different remote sensing sensors corresponding to different water bodies of the world. This will provide great help in establishing regional or unified TSS semi-empirical remote sensing model of estuaries and coasts in the world.

# Code and data availability

The LEDAPS code used for atmospheric correction and all the remote sensing imageries are fully available in the supplement to the article.

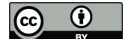



# Acknowledgments

This research was supported by Science & Technology Plan Project of Guangdong Province (2014A020216027, 2016A020223011), Technology transformation project of Zhongshan City-Guangdong Academy of Sciences (2016G1FC0017) and the Water Conservancy Science & Technology Innovation Project (2011-20). The authors would like to thank USGS for providing the Landsat imagery. Thanks are also given to the International Scientific Data Service Platform, Computer Network Information Center, Chinese Academy of Sciences for providing gap filled ETM+ SLC-off data (http://datamirror.csdb.cn).

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



# Tables

**Table 1.** Information about the in-situ data

| Location | Date | Samples | Measurements | Number of synchronous samples with satellite |
|----------|------|---------|--------------|----------------------------------------------|
| Region a | Dec 3, 2010 | 10 | Reflectance, TSS | No |
| | Jan 13~14, 2013 | 22 | Reflectance, TSS | No |
| Region b | Dec 6, 2013 | 11 | Reflectance, TSS | 7, Landsat OLI |
| | Dec 19, 2006 | 5 | Reflectance, TSS | No |
| Region c | Dec 21, 2006 | 18 | 8 samples with Reflectance and TSS; 10 samples with TSS only | 13, EO-1 Hyperion |
| | Dec 27, 2007 | 8 | Reflectance, TSS | No |
| | Nov 2, 2012 | 9 | Reflectance, TSS | 6, Landsat ETM+ |
| Region d | Dec 1, 2013 | 12 | Reflectance, TSS | 9, Landsat OLI |
| Region e | Oct 14~15, 2009 | 34 | Reflectance, TSS | No |

**Table 2.** Review of previous TSS or Turbidity retrieval models using Landsat imagery.

| Data | Model | Study area | Reference |
|------|-------|------------|-----------|
| TM 2 and 4 | TSS=29.022*exp(0.0335*(B4/B2)) | Gironde and Loire Estuaries | Doxaran et al. (2003) |
| MSS 5 and 6 | Ln(TSS)= 1.4*(B5/B6)$^2$-6.2*(B5/B6) +10.8 | the Bay of Fundy and the Beaufort Sea | Topliss et al. (1990) |
| TM 1, 3, 4 | Turbidity=11.31*(B4/B1)-2.03*B3-16.42 | Chagan Lake | Song et al. (2011) |
| TM 1, 2, 3 or 4 | Turbidity=16.1*B4-12.7<br>Turbidity=10*B3-24.8<br>Turbidity=19*B1-97.9<br>Turbidity=6.4*B2-28 | Nebraska Sand Hills Lakes | Fraser. (1998) |
| TM 3 | TSS=69.39*B3-201 | Ganges and Brahmaputra | Islam et al. (2001) |



| Sensor | Equation | Site | Reference |
|---|---|---|---|
| MSS 1 and 2 | Ln(TSS)= 2.71*(B1/B2)$^2$-9.21*(B1/B2) +8.45 | Rivers Enid Reservoir in North Central Mississippi | Ritchie and Cooper. (1991) |
| TM 3 | Log(TSS)= 0.098*B3+0.334 | Delaware Bay | Keiner and Yan. (1998) |
| TM 2 and 3 | TSS=0.7581*exp(61.683*(B2+B3)/2) | southern Frisian lakes | Dekkera et al. (2001) |
| TM 1 and 3 | TSS=0.0167*exp(12.3*B3/B1) | an embayment of Lake Michigan | Lathrop et al. (1991) |
| TM 3 and ETM+ 3 | Log(TSS)= 44.072*B3 +0.1591 | Yellow River estuary | Zhang et al. (2014) |
| TM 3 and 4 | TSS=2.19*exp(21.965*B3) <br> TSS=-9275.78*(B3)$^3$+8623.19*(B3)$^2$ -810.04*B3+23.44 <br> TSS=5829.8*(B3-B4)$^3$+4165.09*(B3-B4)$^2$ -189.88*(B3-B4)+5.43 <br> TSS=3.411*exp(21.998*(B3-B4)) | Poyang Lake | Wu et al. (2013) |
| OLI 2, 3 and 8 | TSS=-191.02*B2+36.8*B3+172.66*B8+4.57 | Xin'anjiang Reservoir | Zhang et al. (2015) |
| TM 2 | B2=0.0044*TSS+2.5226 | Bhopal Upper Lake | Rao et al. (2009) |
| TM 2 and 3 | Log(TSS)= 6.2244*(B2+B3)/B2*B3+0.892 | Yangtze estuary | Li et al. (2010) |
| TM 3 | TSS=0.543*B3-7.102 | Beysehir Lake | Nas et al. (2010) |
| TM 4 | TSS=229457.695*(B4)$^2$+146.462*B4+5.701 | Bohai gulf | Chen et al. (2014) |



**Table 3.** The comparison of calibration and validation accuracy of several best TSS retrieval models

| Model form | | Calibration $R^2$ | Validation (RMSE(mg/L), MRE) | | |
|---|---|---|---|---|---|
| | | | Whole | Low range | High range |
| Chen et al. (2014) | | 0.7842 | 35.7, 144.2% | 18.35, 215.58% | 53.56, 23.5% |
| Li et al. (2010) | | 0.6167 | 69.3, 45% | 5.66, 52.6% | 113.48, 32.1% |
| Zhang et al. (2014) | | 0.5804 | 82.8, 48% | 6.56, 53.9% | 135.54, 38.15% |
| Lathrop et al. (1991) | | 0.6661 | 50.3 39.4% | 6.12, 44.6% | 82.09, 30.57% |
| Ritchie and Cooper. (1991) | | 0.6983 | 68.7, 41.3% | 7.24, 44% | 112.32, 36.6% |
| QRLTSS | OLI | 0.7181 | 21.5, 27.2% | 3.5, 31.1% | 35.1, 20.7% |
| model | ETM+ | 0.708 | 25, 32.5% | 4.6, 38.3% | 40.7, 20.3% |
| based on | TM | 0.7079 | 24.9, 31.5% | 4, 35.3% | 40.6, 25.1% |

**Table 4.** The performance and the vertex of quadratic model of different sensors

| | TM | ETM+ | OLI | MODIS |
|---|---|---|---|---|
| | Red band, Near infrared band | Red band, Near infrared band | Red band, Near infrared band | Red band, Near infrared band |
| Wavelength (nm) | 630-690, 760-900 | 630-690, 775-900 | 630-680, 845-885 | 620-670, 841-874 |
| Spatial resolution (m) | 30 | 30 | 30 | 250 |
| Radiometric resolution (bit) | 8 | 8 | 12 | 12 |
| Signal/noise (dB. Specified level of high ) | 140, 244 | 140, 244 | 340, 460 | 128, 201 |
| The vertex of quadratic model | 0.031, 32.2 mg/L (This study) | 0.031, 32.2 mg/L (This study) | 0.032, 36.1 mg/L (This study) | 0.025, 31 mg/L (Chen et al., 2015b) |



# Figures

![Figure 1 maps]

**Figure 1.** Study areas and locations of in situ data. a: Xuwen coast; b: Moyangjiang River Estuary; c: Pearl River Estuary; d: Hanjiang River Estuary; e: Yangtze River Estuary.



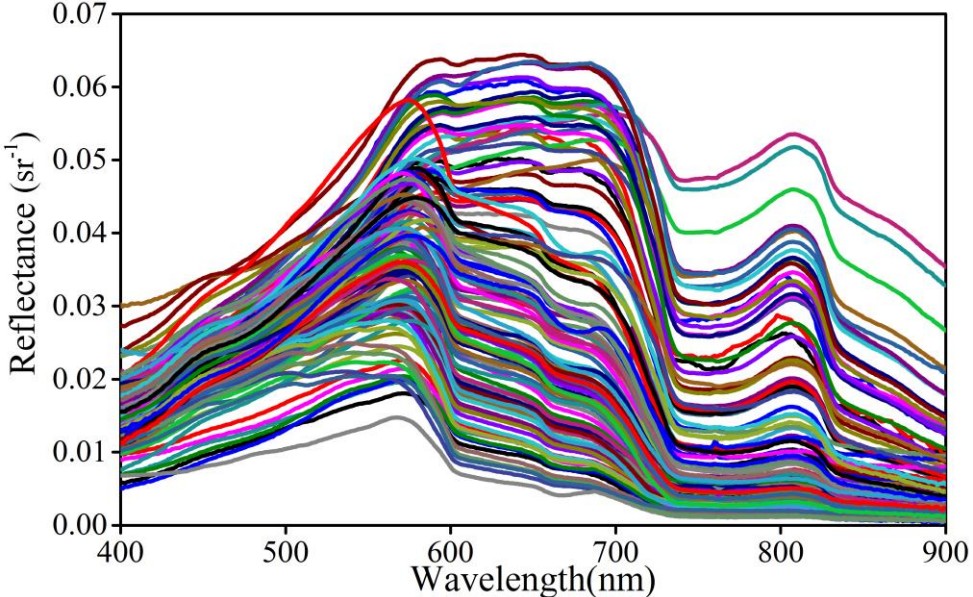

**Figure 2.** 119 spectra were collected from study areas by ASD



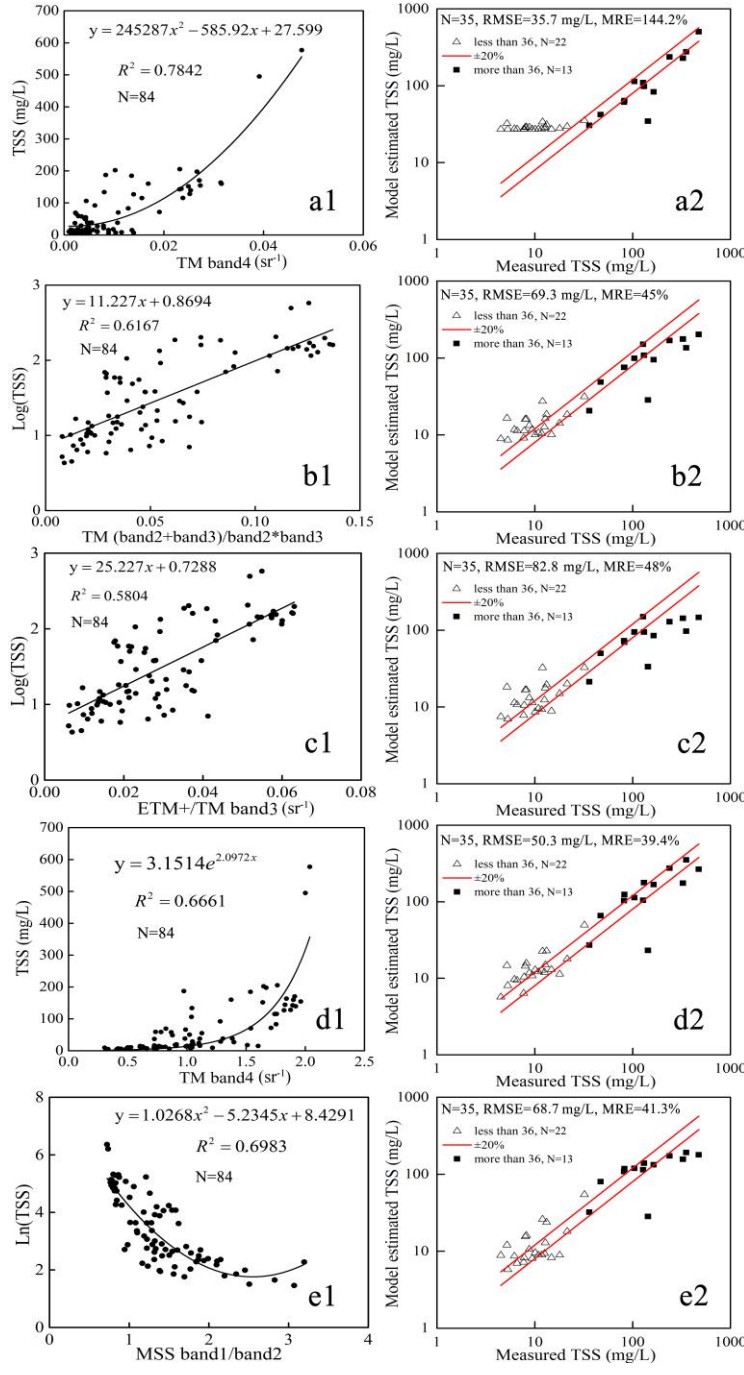

**Figure 3.** The recalibration and validation of previous five TSS retrieval models based on 119 in situ samples. The models were developed by a (Chen et al., 2014), b (Li et al., 2010), c (Zhang et al., 2014), d (Lathrop et al., 1991), e (Ritchie and Cooper., 1991), respectively.



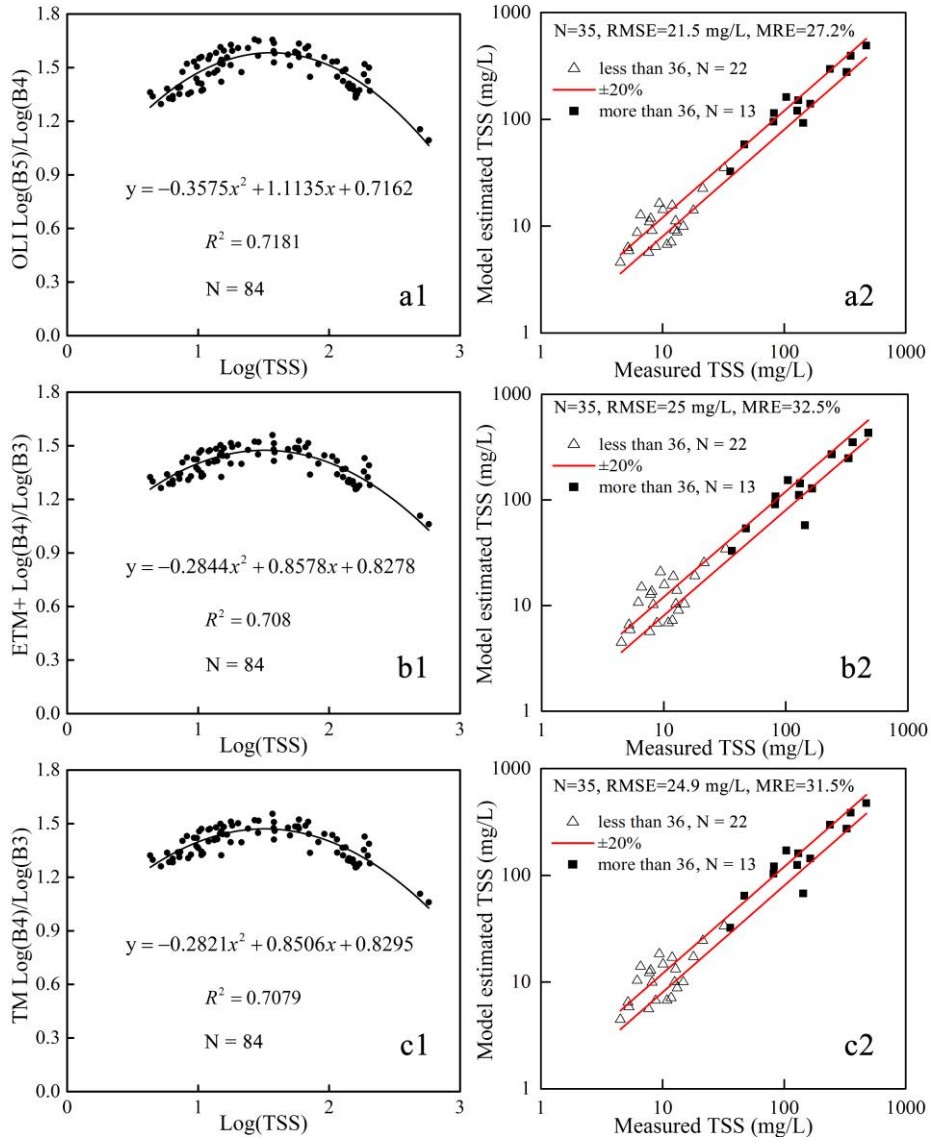

**Figure 4.** The calibration and validation results of TSS retrieval models: based on 119 in situ data (a) OLI, (b) ETM+ and (c) TM.





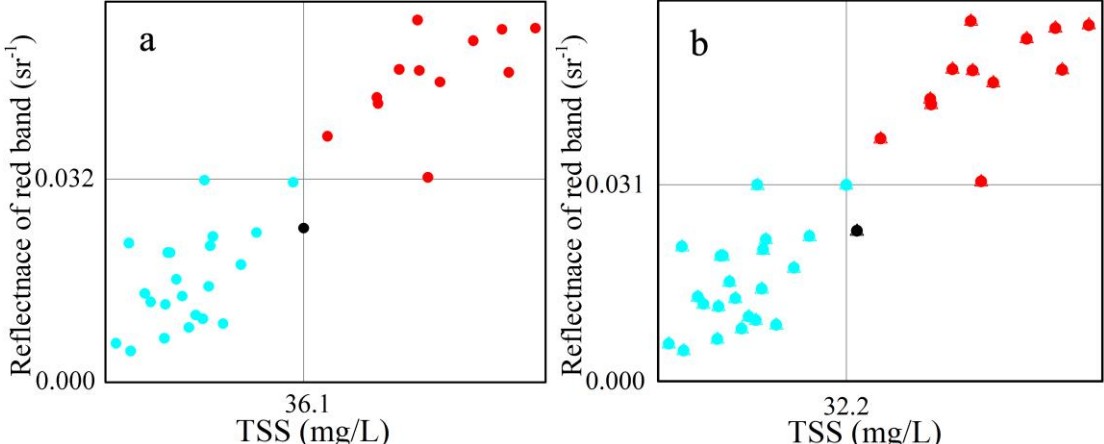

**Figure 5.** Relationship between the Landsat red band reflectance and corresponding TSS concentration. a: OLI sensor, b: ETM+ and TM sensors.

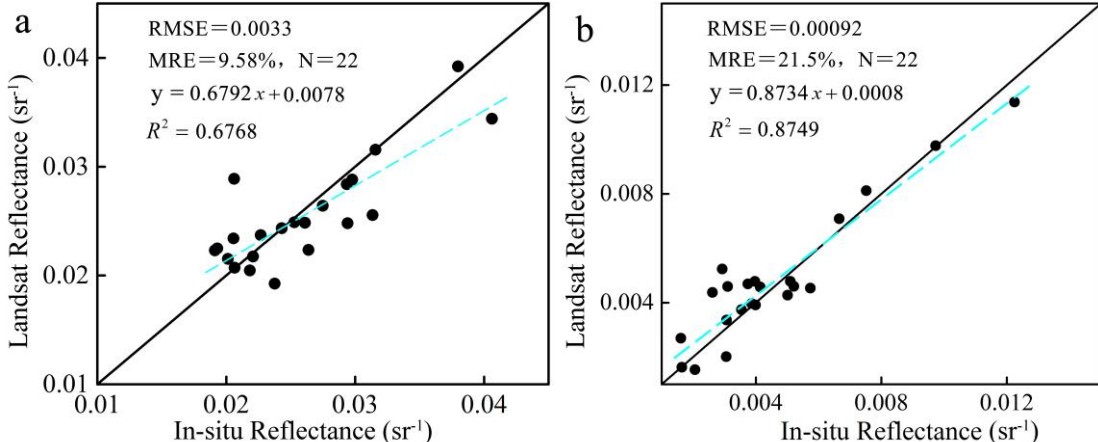

**Figure 6.** Scatterplot of Landsat measured reflectance versus in-situ reflectance. The former is calculated by averaging over a box of 3x3 pixels centered samples. a: red band, b: near infrared band.





**Figure 7.** Estimated TSS concentrations based on QRLTSS model in Moyangjiang River Estuary at 11:00 (Beijing time, OLI), on December 6, 2013 (a), Pearl River Estuary at 10:48 (Beijing time, ETM+), on November 2, 2012 (b), Hanjiang River Estuary at 10:41 (Beijing time, OLI), on December 1, 2013 (c), and comparison between the in-situ measured and Landsat imagery inversed TSS concentrations of three estuaries (d). Color scale is the legend of the TSS concentrations, in mg/L.



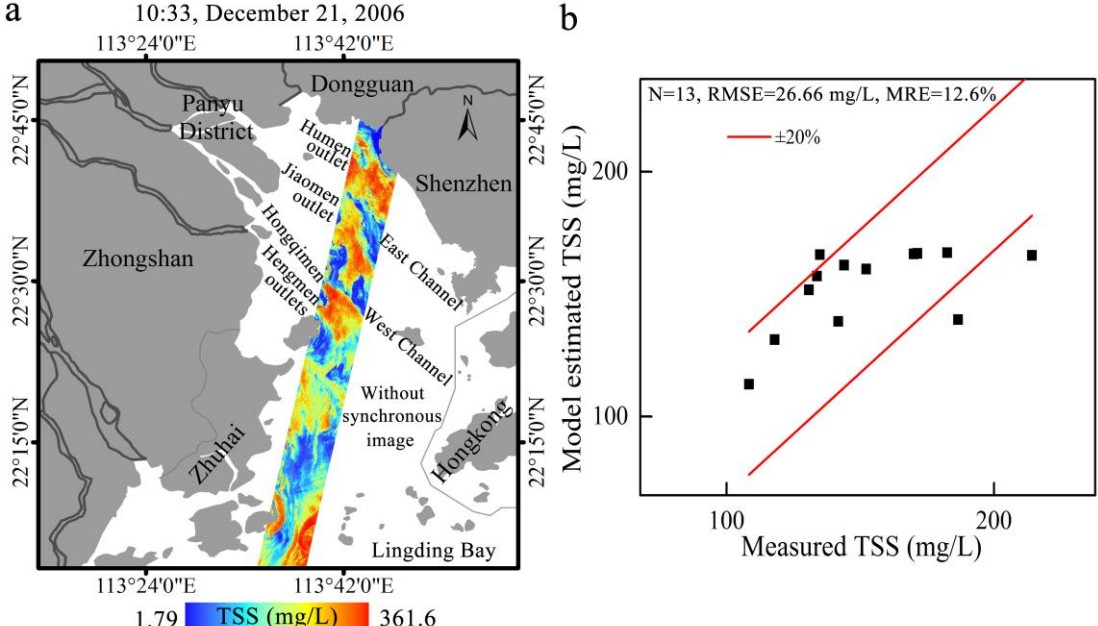

**Figure 8.** Estimated TSS concentrations based on QRLTSS model from EO-1 Hyperion imagery in Pearl River Estuary at 10:33 (Beijing time), on December 21, 2006 (a) and comparison between the in-situ measured and EO-1 Hyperion imagery inversed TSS concentrations (b). Color scale is the legend of the TSS concentrations, in mg/L.