# Peer review of "A Landsat-based model for retrieving total suspended solids concentration of estuaries and coasts in China"

_Geoscientific Model Development, 2016_

## Referee Comment (RC1) · O. Montanher (Referee) · 7 Jun 2017

General comments

Writing: Although my English is not very good, I've noted some writing errors along the paper. Although I've marked a few points, I strongly suggest an additional professional overhaul.

About the title: The title does not represents precisely the content of the paper. I think that it needs to include the geographic specificity of the model ("only" in China). This specification does not denigrate the research relevance, because the coastal environ-

ments of China are giant, have great social importance and have a rich remote sensing TSS modelling background.

Page 1 (abstract): The abstract contains a great number of details concerning the validation and modelling results. However, a first read (without reading the whole paper) does not provided an understanding of the general strategy of modelling. For example, in the line 15 is written N=129, while in lines 18-20 the N (model) is 84 and N (validation) is 35 (84+35 = 119?). After this, another model is mentioned (lines 27-28). A reorganization of the abstract is needed, taking into account the explanation of the modelling, not only the specific indexes as RMSE and $R^2$.

A very important source of error in estimating water components concentration and/or water quality parameters is the sun glint effect. However, there are simple strategies to remove the glint which are based on the SWIR bands. These bands are available for many Landsat sensors. I suggest reading the following papers:

HEDLEY, J. D.; HARBORNE, A. R.; MUMBY, P. J. Simple and robust removal of sun glint for mapping shallow-water benthos. International Journal of Remote Sensing, v. 26, p. 2107 −2112, 2005.

HOCHBERG, E. J.; ANDREFOUET, S.; TYLER, M. R. Sea surface correction of high spatial resolution IKONOS images to improve bottom mapping in near-shore environments. IEEE Transactions on Geoscience and Remote Sensing, v. 41, p. 1724 – 1729, 2003.

KAY, S.; HEDLEY, J. D.; LAVENDER, S. Sun Glint Correction of High and Low Spatial Resolution Images of Aquatic Scenes: a Review of Methods for Visible and Near-Infrared Wavelengths. Remote Sensing, v. 1, n. 4, p. 697-730, 2009.

Wang, M., 2007. Remote sensing of the ocean contributions from ultraviolet to near-infrared using the shortwave infrared bands: simulations. Applied Optics 46,1535–1547.

Wang, J.J., Lu, X.X., Liew, S.C., Zhou, Y., 2009. Retrieval of suspended sediment concentrations in large turbid rivers using Landsat ETM+: an example from the Yangtze River, China. Earth Surface Processes and Landforms 34, 1082–1092.

Following these references, I've used the band 5 (Landsat 5, TM) as a proxy of the Sun glint effect in modelling TSS in the Amazon basin (with co-authors, of course):

Montanher, O. C.; Novo, E. M. L. M.; Barbosa, C. C. F.; Renno, C. D.; Silva, T. S. F. Empirical models for estimating the suspended sediment concentration in Amazonian white water rivers using Landsat/TM. International Journal of Applied Earth Observation and Geoinformation, v. 29, p. 67-77, 2014.

Removing Sun glint effect might improve significantly your results. For example, even the reflectance threshold (0.032 in the red band,) could be affected by Sun glint, resulting in wrong choices (see equation 5 and page 12, lines 23-27). At this stage, performing the sun glint removal means remake all the work... So, this option could be a future strategy for your work.

Specific comments Page 1 - line 21: is there a statistical significant difference between the $R^2$ values? I think that the $R^2$ values difference does not justify: "The QRLTSS model based on Landsat OLI is better than TM and ETM+ ... because of the optimization of OLI sensor's design." A way to verify that may be by means of a statistical test. Page 9 – lines 9-16: This part of the manuscript is a methodology step, including the equation 3. Page 17 – line 13: this sentence is excessively repetitive along the whole paper: "a quadratic model using the ratio of logarithmic transformation of red band and near infrared band and logarithmic transformation of TSS concentration (QRLTSS) for estimating TSS concentration". If the acronym was proposed, would be better use it along the paper. Page 18 – line 7: the 30 m resolution only began in 1982, with Landsat 4. Rewrite to take into account the 80 m MSS.

Technical corrections Page 4 - line 26: Based "on" the above analysis... Page 4 - line 28: Rewrite the beginning of the sentence: "And we focus on..." Figure 1: The map of

China should be improved and noted as "a". Add a scale, coordinates, etc.. Page 5 – lines 13-14: ton/year (or tons per year..), on average. $3 \cdot 10^4$ instead of using 3*104. Reorganize these notations along the whole paper. Page 5 – line 21: 199 km and $6 \cdot 10^3$ km$^2$ Page 6 – lines 22-26: suggestion: include this information in a table. Page 7: "TM, ETM+ and OLI sensors onboard the Landsat 5, 7 and 8 satellites, respectively, have..."

---

## Editor Comment (EC1) · B. Jackson (Editor) · 30 Jun 2017

We have really struggled finding enough reviewers who have not only the required technical and research competency to evaluate this manuscript but also the time to carry out a review. My apologies to the authors the process of review has therefore been slower than is normal, and thanks for providing your response to the existing published review.

A further review is still underway and I will evaluate the manuscript with recommendations as a matter of urgency once this is received.

---

## Author Comment (AC1) · 30 Jun 2017

Dear Mr. Montanher,

Many thanks for your valuable comments. Following your thoughtful and helpful suggestions, we have revised the manuscript carefully. The list below is the response.

*General comments*
*Writing: Although my English is not very good, I've noted some writing errors along the paper. Although I've marked a few points, I strongly suggest an additional professional overhaul.*
**Response:**

Thanks for correction of our manuscript in language and grammar. Following your suggestions, we have also asked for two associated professors of USA helping us to pick up writing errors and revise our manuscript very carefully. All the revising have been marked in red in the marked-up mode (Page 1, lines 1-2; Page 2, line 6; Page 3, lines 1-4, 26; Page 5, lines 7, 11, 20, 25; Page 6, lines 8, 16; Page 12, line 21; Page 15, lines 3, 14, 21; Page 16, line 13).

*About the title: The title does not represents precisely the content of the paper. I think that it needs to include the geographic specificity of the model ("only" in China). This specification does not denigrate the research relevance, because the coastal environments of China are giant, have great social importance and have a rich remote sensing TSS modelling background.*
**Response:**

Dear Mr. Montanher, following your suggestion, we added the geographic specification into the title (Page 1, lines 15, 19-20). We agree that the title "A Landsat-based model for retrieving total suspended solids concentration of estuaries and coasts in China" represents the content of the paper more precisely than the one before.

*Page 1 (abstract): The abstract contains a great number of details concerning the validation and modelling results. However, a first read (without reading the whole paper) does not provided an understanding of the general strategy of modelling. For example, in the line 15 is written N=129, while in lines 18-20 the N (model) is 84 and N (validation) is 35 (84+35 = 119?). After this, another model is mentioned (lines 27-28). A reorganization of the abstract is needed, taking into account the explanation of the modelling, not only the specific indexes as RMSE and $R^2$.*
**Response:**

We did the revision that there are **129** in total in-situ samples were collected from the study areas (Page 6, line 23). Among them, there are **119** in-situ samples with field spectral measurements and synchronous water samples (TSS concentration) (Page 7, lines 3-4. Table 1). **Another ten** in-situ samples have TSS concentration **only** while EO-1 Hyperion overpasses (Page 7, lines 4-5. Table 1). So, the **84 and 35** in-situ samples were used to calibrate and validate the

QRLTSS model respectively. Another **ten** in-situ samples of TSS were **only** used to validate the accuracy of the QRLTSS model **for remote sensing inversion of Hyperion image**. The line 27-28 mentioned was the threshold method of solving the two-valued issue of QRLTSS model.

*A very important source of error in estimating water components concentration and/or water quality parameters is the sun glint effect. However, there are simple strategies to remove the glint which are based on the SWIR bands. These bands are available for many Landsat sensors. I suggest reading the following papers:*
*HEDLEY, J. D.; HARBORNE, A. R.; MUMBY, P. J. Simple and robust removal of sun glint for mapping shallow-water benthos. International Journal of Remote Sensing, v. 26, p. 2107-2112, 2005.*
*HOCHBERG, E. J.; ANDREFOUET, S.; TYLER, M. R. Sea surface correction of high spatial resolution IKONOS images to improve bottom mapping in near-shore environments. IEEE Transactions on Geoscience and Remote Sensing, v. 41, p. 1724 – 1729, 2003.*
*KAY, S.; HEDLEY, J. D.; LAVENDER, S. Sun Glint Correction of High and Low Spatial Resolution Images of Aquatic Scenes: a Review of Methods for Visible and Near-Infrared Wavelengths. Remote Sensing, v. 1, n. 4, p. 697-730, 2009.*
*Wang, M., 2007. Remote sensing of the ocean contributions from ultraviolet to near infrared using the shortwave infrared bands: simulations. Applied Optics 46, 1535–1547.*
*Wang, J.J., Lu, X.X., Liew, S.C., Zhou, Y., 2009. Retrieval of suspended sediment concentrations in large turbid rivers using Landsat ETM+: an example from the Yangtze River, China. Earth Surface Processes and Landforms 34, 1082–1092.*

*Following these references, I've used the band 5 (Landsat 5, TM) as a proxy of the Sun glint effect in modelling TSS in the Amazon basin (with co-authors, of course):*
*Montanher, O. C.; Novo, E. M. L. M.; Barbosa, C. C. F.; Renno, C. D.; Silva, T. S. F. Empirical models for estimating the suspended sediment concentration in Amazonian white water rivers using Landsat/TM. International Journal of Applied Earth Observation and Geoinformation, v. 29, p. 67-77, 2014.*

*Removing Sun glint effect might improve significantly your results. For example, even the reflectance threshold (0.032 in the red band,) could be affected by Sun glint, resulting in wrong choices (see equation 5 and page 12, lines 23-27). At this stage, performing the sun glint removal means remake all the work… So, this option could be a future strategy for your work.*
**Response:**

Yes, the sun glint effect is indeed an important source of error in estimating water components concentration. However, the 6S atmospheric correction model this study used also corrects the skylight reflection (Sun and sky glint) following the Snell-Fresnel laws, environmental effects and directional target effects (Doxarana et al. 2002), and the Fresnel reflection is partially reduced by the presence of land in estuarial and coastal waters (Vidot and

Santer 2005). Actually, we **also agree** that there are still partial residues of glint effect remained. As we all know, each of atmospheric correction methods has its advantages and disadvantages. It is an interesting question and deserves our further study according to your suggestion. We have made partial revision in section 2.3 (Atmospheric correction method) for the clarity (Page 8, lines 9-12). The following references have been added.

Doxaran, D., Froidefond, J.M., Lavender, S., & Castaing, P.: Spectral signature of highly turbid waters Application with SPOT data to quantify suspended particulate matter concentrations. Remote Sensing of Environment, 81, 149-161, doi.org/10.1016/S0034-4257(01)00341-8, 2002.

HEDLEY, J.D., HARBORNE, A.R., MUMBY, P.J.: Simple and robust removal of sun glint for mapping shallow-water benthos. International Journal of Remote Sensing, 26(10), 2107-2112, DOI: 10.1080/01431160500034086, 2005.

Hochberg, E.J., Andréfouët, S., Tyler, M.R.: Sea surface correction of high spatial resolution IKONOS images to improve bottom mapping in near-shore environments. IEEE Transactions on Geoscience and Remote Sensing, 41(7), 1724-1729, DOI: 10.1109/TGRS.2003.815408, 2003.

Montanher, O.C., Novo, E.M.L.M., Barbosa, C.C.F., Renno, C.D., Silva, T.S.F.: Empirical models for estimating the suspended sediment concentration in Amazonian white water rivers using Landsat/TM. International Journal of Applied Earth Observation and Geoinformation, 29, 67-77, DOI: 10.1016/j.jag.2014.01.001, 2014.

Vidot, J., & Santer, R.: Atmospheric correction for inland waters-application to SeaWiFS. International Journal of Remote Sensing, 26, 3663-3682, DOI: 10.1080/01431160500034029, 2005.

*Specific comments Page 1 - line 21: is there a statistical significant difference between the $R^2$ values? I think that the $R^2$ values difference does not justify: "The QRLTSS model based on Landsat OLI is better than TM and ETM+ ... because of the optimization of OLI sensor's design." A way to verify that may be by means of a statistical test. Page 9 – lines 9-16: This part of the manuscript is a methodology step, including the equation 3. Page 17 – line 13: this sentence is excessively repetitive along the whole paper: "a quadratic model using the ratio of logarithmic transformation of red band and near infrared band and logarithmic transformation of TSS concentration (QRLTSS) for estimating TSS concentration". If the acronym was proposed, would be better use it along the paper. Page 18 – line 7: the 30 m resolution only began in 1982, with Landsat4. Rewrite to take into account the 80 m MSS.*

**Response:**

In fact, the difference between the $R^2$ values (calibration accuracy) of QRLTSS models based on OLI, ETM+ and TM is very small with the statistical significances (P-value < 0.001). The validation accuracy of QRLTSS models based on OLI is also a little better than ETM+ and TM (RMSE: 21.5 mg/L vs 25 mg/L and 24.9 mg/L, MRE: 27.2% vs 32.5% and 31.5%). We come to the conclusion of OLI better based on the integrated consideration of calibration and validation accuracy of the model, and the performance of different sensors. We have revised the corresponding statement to make it more clear (Page 1, lines 20-22).

The contents of the Part "Band response function application of Landsat sensors for field spectra" had been moved to Section 2.5 (Pages 9-10).

We have checked the manuscript and used the acronym (QRLTSS) along the paper to represent *"a quadratic model using the ratio of logarithmic transformation of red band and near infrared band and logarithmic transformation of TSS concentration for estimating TSS concentration"* (Page 17, lines 17 and 21).

On the beginning of 30 m resolution, we have modified the corresponding sentence as "Landsat imageries could be one of the best choices in terms of the availability of data source for remote sensing of TSS in estuaries and coasts, considering the spatial resolution and acquirement of long time series (30m TM/ETM+/OLI beginning in 1982, 80m MSS since 1972)" (Page 18, lines 2-3).

*Technical corrections Page 4 - line 26: Based "on" the above analysis…Page 4 – line 28: Rewrite the beginning of the sentence: "And we focus on…" Figure 1: The map of China should be improved and noted as "a". Add a scale, coordinates, etc. Page 5 –lines 13-14: ton/year (or tons per year.), on average. $3 \cdot 10^4$ instead of using $3*10^4$. Reorganize these notations along the whole paper. Page 5 – line 21: 199 km and 6.103 km² Page 6 – lines 22-26: suggestion: include this information in a table. Page 7: "TM, ETM+ and OLI sensors onboard the Landsat 5, 7 and 8 satellites, respectively, have…"*

**Response:**

We have revised the statements (Page 4, line 26) that you pointed out and checked the whole manuscript carefully (Page 4, line 24). The sentence "And we focus on the models examination of multiple bands combination, which belong to non-monotonic function" have been modified as "We focus on the models of multiple bands combination, and the form of models belongs to non-monotonic function." (Page 4, lines 26-27).

Following your suggestions, the associated hydrologic features (length, drainage area, mean surface runoff and sediment discharge) of five estuaries or coasts have been added into Table 1 (Page 25). The map of China was noted as "a", and added a scale, a north arrow, coordinates as well (Page 29). The revised Table 1 and Figure 1 have also been shown as follows

Once again, thank you very much for your valuable comments and suggestions.

**The revised Table 1.** Information about the study areas and in-situ data

| Location | Hydrologic features (length, drainage area, mean surface runoff and sediment discharge) | Date | Samples | Measurements | Number of synchronous samples with satellite |
|---|---|---|---|---|---|
| Region I | $-$, $-$ $8.6•10^8 m^3/year$, $3•10^4$ ton/year | Dec 3, 2010 | 10 | Reflectance, TSS | No |
| | | Jan 13~14, 2013 | 22 | Reflectance, TSS | No |
| Region II | 199 km, $6•10^3 km^2$, $8.21•10^9 m^3/year$, $3.27•10^5$ ton/year | Dec 6, 2013 | 11 | Reflectance, TSS | 7, OLI |
| Region III | 2320 km, $4.53•10^5 km^2$, $3.26•10^{11} m^3/year$, $7.53•10^7$ ton/year | Dec 19, 2006 | 5 | Reflectance, TSS | No |
| | | Dec 21, 2006 | 18 | 8 samples with Reflectance and TSS; 10 samples with TSS only | 13, Hyperion |
| | | Dec 27, 2007 | 8 | Reflectance, TSS | No |
| | | Nov 2, 2012 | 9 | Reflectance, TSS | 6, ETM+ |
| Region IV | 470 km, $3.01•10^4 km^2$, $2.45•10^{10} m^3/year$, $6.93•10^6$ ton/year | Dec 1, 2013 | 12 | Reflectance, TSS | 9, OLI |
| Region V | 6280 km, $1.8•10^6 km^2$, $9.2•10^{11} m^3/year$, $4.8•10^8$ ton/year | Oct 14~15, 2009 | 34 | Reflectance, TSS | No |

[Figure]

**The revised Figure 1.** Study areas and locations of in situ data. b: Xuwen coast; c: Moyangjiang River Estuary; d: Pearl River Estuary; e: Hanjiang River Estuary; f: Yangtze River Estuary.

**A Landsat-based model for retrieving total suspended solids concentration of estuaries and coasts in China**

Chongyang Wang[1,2,3], Shuisen Chen[2], Dan Li[2], Wei Liu[1,2,3], Ji Yang[1,2,3], Danni Wang[4]

[1] Guangzhou Institute of Geochemistry, Chinese Academy of Sciences, Guangzhou 510640, China

5    [2] Guangzhou Institute of Geography, Guangzhou 510070, China

[3] University of Chinese Academy of Sciences, Beijing 100049, China

[4] Department of Resources and the Urban Planning, Xin Hua College of Sun Yat-Sen University, Guangzhou 510520, China

*Correspondence to*: Shuisen Chen (css@gdas.ac.cn)

10    **Abstract.** Retrieving total suspended solids (TSS) concentration accurately is essential for sustainable management of estuaries and coasts, which plays a key role in the interaction of hydrosphere, pedosphere and atmosphere. Although many TSS retrieval models have been published, the general inversion method that is applicable to different field conditions is still under research. In order to obtain a TSS remote sensing model that is suitable for estimating the TSS concentrations with wide range in estuaries and coasts by Landsat imageries, this study recalibrated and validated a

15    number of regression-techniques-based TSS retrieval models using 119 in-situ samples collected from five regions of China during the period of 2006-2013. It was found that the adjusted Quadratic model using the Ratio of Logarithmic transformation of red band and near infrared band and logarithmic transformation of TSS concentration (QRLTSS) works well and shows a relatively satisfactory performance. The adjusted QRLTSS model based on Landsat TM, ETM+ and OLI sensors explained about 72% of the TSS concentration variation (TSS: 4.3-577.2 mg/L, N=84. P-value

20    < 0.001) and had an acceptable validation accuracy (TSS: 4.5-474 mg/L, RMSE: about 23 mg/L, N=35). The accuracy of QRLTSS model based on Landsat OLI is a little higher than TM and ETM+ due to 
[revised manuscript text omitted]

---

## Author Comment (AC2) · 30 Jun 2017

The authors would like to thank Mr. Montanher for the efforts in helping us to improve the manuscript. Our full response is included as a pdf supplement.

Please also note the supplement to this comment: https://www.geosci-model-dev-discuss.net/gmd-2016-297/gmd-2016-297-AC2-supplement.pdf

---

## Referee Comment (RC2) · Anonymous Referee #2 · 21 Aug 2017

General comments

The authors have provided a potential novel methodology to estimate total suspended solids concentrations of estuaries and coastal areas of China. However, we have a number of concerns about the approach that should be considered before being allowed to advance further through the review process. Unfortunately, there are numerous grammatical and structural errors throughout the document which has made comprehension difficult in some places. The flow between sentences, and paragraphs could be improved, with better linkages between ideas, concepts and explanations. Before the next submission, we strongly suggest having a native English speaker review

and correct the manuscript.

While the background is quite easy to follow, the rationale for this study and what it is aiming to achieve could be better articulated. It is not clear in the abstract that the authors are evaluating a model they themselves have developed nor does it mention the novelty of their approach. It reads more like they are just testing an existing method. They also mention the 'adjusted' QRLTSS model (line 18), but I am not sure how this differs to the QRLTSS model mentioned in the previous sentence (line 16).

Of particular concern is the sampling method for the in-situ data. There is a lack of information about the time and depth of each sample, and then how this corresponds to the timing of the satellite image used in the analysis. Samples were taken between 10 am to 5 pm, during which it is likely tides may have significantly altered turbidity and TSS concentrations. But the tidal effect on turbidity has not been mentioned. Further some analysis of the time of the satellite image and the sample should be undertaken – a longer length of time between the two may impact the accuracy of results. Finally, we would like some justification for the location of the samples taken. Sampling locations tend to be in areas which have similar turbidity characteristics and do not represent the wide variability of tss/turbidity within the individual estuaries and coastal areas under investigation. As a result, we have concerns about the validity of the approach for extrapolating over larger areas in which there will be a large amount of variability in TSS concentrations. We would recommend that the limitations of the work are discussed more thoroughly with this in mind. Alternatively, considerably more samples would need to be collected that cover a wider area of the estuaries and coastal areas under investigation.

Specific comments

Abstract – states 129 samples were used but in the body of the text 119 were used in total.

Page 7, line 7-10. Please explain ETM+, TM and OLI in more detail and how they differ.

Page 8 Section 2.3. It is not clear why atmospheric correction is needed, nor why the 6S method has been chosen above other methods.

Page 9, Section 2.4. There is no justification for why all three assessment methods are needed or why they are chosen. Please clarify.

Page 9, section 3.1. It needs to be clearer why the reflectance in the red band and near infrared band is only used.

Page 9, Eqn3 – r is not defined in the numerator.

There is no discussion of the impact of sun glint and marine vessel contamination in the area (especially in the Pearl River estuary) – how is this dealt with?

The authors mention that they review and analyse 20 models (section 1) but only 5 models are discussed later (section 3.1), with little discussion about why only 5 were selected.

Depth of samples and time of each sample should be mentioned.

Some samples were taken from clear water (e.g. like the ones from Yangtze river estuary) which impacts reflectance due to higher amounts of water vapour and aerosols. This would need to be accounted for.

Some samples were taken from narrow canals with a width a bit bigger than TM spatial resolution. There would be an effect from the surrounding land on the reflectance of this water. This effect should be discussed.

There was minimal variability in sampling locations in each estuary/coastal area. Samples were taken either in areas which were turbid or clear, despite there being wider ranging turbidity in the area.

The selection of sampling locations should be discussed, with comment on why the sampling locations tended to be clustered in this way.

It is better to mention that root mean square of Trimble global positioning system is related to real time or post processed accuracy (page 6 line 21)

The process of matching image and sample points should be discussed.

Section 3.2, page 12 line 12. I am not clear how the vertex values have been calculated. This should be clarified.

Page 18, line 1. New results have been introduced in the summary and conclusions. These should be in the body of the text or in figures first.

Technical corrections

There are numerous grammatical and structural issues that should be addressed. This manuscript would benefit from editing and review from a native English speaker to improve structure, flow and readability.

Section 2.2, I would recommend the dates being in day month year order to make them easier to read/follow.

Sentences should not start with the word 'And'

The writing is sometimes too conversational. For example, page 4 line 20 "What's more" could be replaced with "furthermore".

Tables 1-3 could be easier to read if each section was more clearly separated. For example, table 1, it is not clear what dates and samples are specific to Region c. Table 2, it is difficult to tell which turbidity models are used for TM 1,3, 4 as opposed to TM, 1,2,3,4. Table 3, I suggest adding a line between the previous models and the ones used in this study. . .etc

Figure 1 should go from a to e rather than e,a,b,c,d, as is currently the case. Need to make it clearer in the caption that the black dots are the sampling locations. Please explain the difference between circles an triangle in figure 1c (page 21).

[Figure]

---

## Author Comment (AC3) · 5 Sep 2017

Dear reviewer,

We appreciate that you provided the valuable comments for the improvement of the manuscript. Following your thoughtful suggestions, we have revised the manuscript very carefully. The lists below are the responses to each comment.

*General comments*

*The authors have provided a potential novel methodology to estimate total suspended solids concentrations of estuaries and coastal areas of China. However, we have a number of concerns about the approach that should be considered before being allowed to advance further through the review process. Unfortunately, there are numerous grammatical and structural errors throughout the document which has made comprehension difficult in some places. The flow between sentences, and paragraphs could be improved, with better linkages between ideas, concepts and explanations. Before the next submission, we strongly suggest having a native English speaker review and correct the manuscript.*

**Response:**

Thanks a lot for the valuable comments about the grammatical and structural issues. Following your suggestions, we have asked for **native speakers** (two associated professors of USA) helping us to pick up grammatical errors and revise the manuscript very carefully. **In addition**, reviewer #1 had also helped to point out main writing errors and structural issues and revised our manuscript carefully. Now the linkage of sentences, paragraphs had been polished and added in many places for the improvement of flow, structure and readability of the manuscript, which is helpful to outline the general modelling strategy more clearly (Page 3, line 19; Page 15, line 14; Page 6, lines 21-25; Page 9, lines 2-4; Section 2.1; Section 2.4; Section 2.5; Section 3.4). The contents of "extraction of water bodies" in Section 3.4 is removed to Section 2.1 (Page 6). The contents of the "Band response function application of Landsat sensors for field spectra" in Section 3.1 have been moved to Section 2.4 (Page 10). All the revising have been marked in red in the marked-up mode (Page 1, lines 1-2, 14-15, 17-23, 25-28; Page 2, lines 1-2, 5-6, 10; Page 3, lines 1-4, 15-16, 18-19, 26; Page 4, lines 20-22, 24-27; Page 5, lines 9-10, 17-18, 22-23; Page 6, lines 5-6, 13-14, 21-25; Page 7; Page 8, lines 1-6, 10-15, 17, 22, 26-27; Page 9, lines 1-3, 7-9, 12-15, 18-21; Page 10, lines 10-11, 14-16; Page 11, lines 6-7; Page 12, lines 2-4; Page 13, lines 14-21, 24; Page 14, lines 18-19; Page 15, lines 6-9, 19-22; Page 16, lines 4-9, 16, 26-27; Page 17, lines 6-7, 17-20, 23-24; Page 18, lines 4-5, 14, 19-23, 27; Page 19, lines 1-4, 6, 9-10, 17, 21-23). We expect that the revised manuscript will meet the demand of Journal.

*While the background is quite easy to follow, the rationale for this study and what it is aiming to achieve could be better articulated. It is not clear in the abstract that the authors are evaluating a model they themselves have developed nor does it mention the novelty of their approach. It reads more like they are just testing an existing method. They also mention the 'adjusted' QRLTSS model (line 18), but I am not sure how this differs to the QRLTSS model mentioned in the previous sentence (line 16).*

**Response:**

We are very sorry that there are controversial words and imprecise statements confusing the understanding of the text. This study did NOT test an existing method. Based on 119 in-situ samples collected in 2006-2013 from five regions of China, this study **developed** the QRLTSS model by **reviewing** a number of Landsat-based TSS retrieval models and **improving** a relatively better one among them. About the "adjusted", we meant improving before. The QRLTSS model mentioned in the manuscript is all the **same one**. We have revised the statements in whole manuscript very carefully to avoid the confusion correspondingly. (Page 1, lines 18-19, 25; Page 2, lines 1-2, 5, 10; Page 12, line 3; Page 18, lines 19, 21, 23, 27; Page 19, lines 6, 10, 17).

*Of particular concern is the sampling method for the in-situ data. There is a lack of information about the time and depth of each sample, and then how this corresponds to the timing of the satellite image used in the analysis. Samples were taken between 10 am to 5 pm, during which it is likely tides may have significantly altered turbidity and TSS concentrations. But the tidal effect on turbidity has not been mentioned. Further some analysis of the time of the satellite image and the sample should be undertaken – a longer length of time between the two may impact the accuracy of results.*

**Response:**

There was a mistake to summarize the information of sampling succinctly in **previous version**. However, we still showed some necessary information about in-situ samples. The water samples and synchronous field spectral measurements were carried out from **10:00 to 15:00** (Page 7, lines 8-9) **rather than 10:00 to 5 pm**. The spectrum were measured based on above-water spectrum measurement method (Tang et al., 2004) (Page 7, lines 12-13). Water samples (about 1.5 L) were collected within the water depth of 1 m. TSS concentrations were measured from water samples by a weighed method (Binding et al., 2012; Caballero et al., 2014) (Page 7, lines 18-19). The sampling method for the in-situ data followed the community accepted standards and was applied widely (Chen et al., 2015; Feng et al., 2014; Zhang et al., 2014) (Page 7, lines 13-14). So, the samples are not difficult to correspond to the timing of the satellite image used in the study.

Following your suggestions, we have revised the relevant statements to make it clearer (Page 7, lines 8-19).

It is right that a longer length of time between sample and satellite image may impact the accuracy of results. Thus, we selected the samples within **two-hour time** window of satellite overpass to validate the accuracy of the QRLTSS model similar to the previous studies (Bailey and Werdell. 2006; Chen et al., 2015; Zhang et al., 2014) (Page 15, lines 16-18; Page 17, lines 25-27).

Indeed, tides would significantly altered turbidity and TSS concentrations as you pointed out. However, the samples were taken mainly from 10:00 to 15:00 (Beijing time) (Page 7, lines 8-9) and the samples used for validation were within **two-hour time** window of satellite overpass in the study (Page 15, lines 16-18; Page 17, lines 25-27). Thus, the tidal effect on turbidity is relatively small (Bailey and Werdell. 2006; Chen et al., 2015). In addition, we had simply analyzed the spatial variation of TSS concentrations with the impact of interaction of tide and runoff in the manuscript (Page 16, lines 10-12, 17-18; Page 17, lines 7-8). The tidal effect on turbidity you pointed out is an interesting question and deserves the further study.

References:

Bailey, S.W., & Werdell, P.J.: A multi-sensor approach for the on-orbit validation of ocean color satellite data products. Remote Sensing of Environment, 102, 12-23, doi.org/10.1016/j.rse.2006.01.015, 2006.

Binding, C.E., Greenberg, T.A., & Bukata, R.P.: An Analysis of MODIS-Derived Algal and Mineral Turbidity in Lake Erie. Journal of Great Lakes Research, 38, 107-116, doi.org/10.1016/j.jglr.2011.12.003, 2012.

Caballero, I., Morris, E.P., Ruiz, J., & Navarro, G.: Assessment of suspended solids in the Guadalquivir estuary using new DEIMOS-1 medium spatial resolution imagery. Remote Sensing of Environment, 146, 148–158, doi.org/10.1016/j.rse.2013.08.047, 2014.

Chen, J., Quan, W., Cui, T., & Song, Q.: Estimation of total suspended matter concentration from MODIS data using a neural network model in the China eastern coastal zone. Estuarine, Coastal and Shelf Science, 155, 104-113, doi.org/10.1016/j.ecss.2015.01.018, 2015.

Feng, L., Hu, C., Chen, X., & Song, Q.: Influence of the Three Gorges Dam on total suspended matters in the Yangtze Estuary and its adjacent coastal waters: Observations from MODIS. Remote Sensing of Environment, 140, 779–788, doi.org/10.1016/j.rse.2013.10.002, 2014.

Tang, J., Tian, G., Wang, X., Wang, X., & Song, Q.: The Methods of Water Spectra Measurement and Analysis: Above-Water Method. Journal of Remote Sensing, 8, 37-44, 2004.

Zhang, M., Dong, Q., Cui, T., Xue, C., & Zhang, S.: Suspended sediment monitoring and assessment for Yellow River estuary from Landsat TM and ETM+ imagery. Remote Sensing of Environment, 146, 136–147, doi.org/10.1016/j.rse.2013.09.033, 2014.

*Finally, we would like some justification for the location of the samples taken. Sampling locations tend to be in areas which have similar turbidity characteristics and do not represent the wide variability of tss/turbidity within the individual estuaries and coastal areas under investigation. As a result, we have concerns about the validity of the approach for extrapolating over larger areas in which there will be a large amount of variability in TSS concentrations. We would recommend that the limitations of the work are discussed more thoroughly with this in mind. Alternatively, considerably more samples would need to be collected that cover a wider area of the estuaries and coastal areas under investigation.*

**Response:**

In-situ samples, covering wide ranging of turbidity, are important to the calibration and variation of TSS model in estuaries and coasts.

In our study areas, Xuwen coast is **a less turbid region** due to less water discharge, sediment load and protection of coral reef (Page 5, lines 7-10). The mean distance of two samples is more than 0.8 km and the farthest distance is about 23.98 km (Page 31, figure 1b), which covers a large area of Xuwen coast (10.77%), including two core areas and a buffer area in Xuwen Coral Reef Reserve. TSS concentrations of the 32 samples ranges from **4.3 mg/L** to **37.8 mg/L** with a mean value of **11.2 mg/L** in Xuwen coast. These samples represent the variability of TSS concentrations in the region well.

TSS concentrations in Moyangjiang River Estuary, Pearl River Estuary and Hanjiang River Estuary of Guangdong Province (Page 31, figure 1c, d, e) were mainly affected by the interaction of runoff and tide. So, the samples in the three estuaries (Page 31, figure 1c, d, e) with the **almost same representativeness** were carried out from the **lower river reaches** to the **outer shelf area** (including shoals, channels, maximum turbidity zones). TSS concentrations of the total of 62 simples (Page 31, figure 1c, d, e) in the three estuaries ranging from **5.2 mg/L** to **220.7 mg/L**. Among them, TSS concentrations of 15 simples ranges from 106 mg/L to 220.7 mg/L with a mean value of **150.9 mg/L**. In addition, the mean distance of two samples and the farthest distance from lower river reaches in the three estuaries are more than 0.7 km and about **15.3 km** in Moyangjiang River Estuary (Page 31, figure 1c), 1.3 km and **27.4 km** in Pearl River Estuary (Page 31, figure 1d) and 0.7 km and **8.7 km** in Hanjiang River Estuary (Page 31, figure 1e), respectively. We believe that the samples can reflect the characteristic of turbidity of estuaries of Guangdong Province well.

Besides, TSS concentrations of the 34 samples in Yangtze Estuary (Page 31, figure 1f) ranges from **38.4 mg/L** to **577.2 mg/L** with a mean value of 178.2 mg/L. The mean distance of two

samples is more than **2.6 km** and the farthest distance from lower river reaches is about **73.32 km** (Page 31, figure 1f). These samples reflect the large variation of TSS concentrations of Yangtze River well.

We also agree that the more samples could represent the high-dynamic TSS concentrations in larger areas better. However, in-situ data collection is very difficult plus the budget issue and weather during the satellite overpass, especially in coastal oceans of cloudy and rainy south China. We collected the samples funded by several projects for many years. Now, we have added the statements to present the representativeness of the location or concentrations of the samples clearer (Pages 7, lines 1-4).

*Specific comments*
*Abstract – states 129 samples were used but in the body of the text 119 were used in total.*
**Response:**

Sorry for the confusing statement. In fact, there are **129** in-situ samples in total collected from the study areas (Page 6, line 27). Among them, there are **119** in-situ samples with field spectral measurements and synchronous water samples (TSS concentrations) (Page 7, lines 8-9. Table 1). **Another ten** in-situ samples **have TSS concentration only** (Page 7, lines 9-11. Table 1). So, the **119** in-situ samples were used to calibrate (N=84) and validate (N=35) the QRLTSS model respectively. The other **ten** in-situ samples of TSS were **only** used to **further validate** the accuracy of the QRLTSS model from remote sensing inversion of Hyperion image. We have revised the statement to make it clearer (Page 1, line 17).

*Page 7, line 7-10. Please explain ETM+, TM and OLI in more detail and how they differ.*
**Response:**

The Landsat Thematic Mapper (**TM**) sensor was carried on Landsat 4 and Landsat 5, and images consist of six spectral bands with a spatial resolution of 30 meters for Bands 1-5 and 7 (**Band 1** - Blue, 0.45 - 0.52 μm; **Band 2** - Green, 0.52 - 0.60 μm; **Band 3** - Red, 0.63 - 0.69 μm; **Band 4** - Near Infrared, 0.76 - 0.90 μm; **Band 5** - Shortwave Infrared (SWIR 1), 1.55 - 1.75 μm and **Band 7** - Shortwave Infrared (SWIR 2), 2.08 - 2.35 μm) and one thermal band (**Band 6** – Thermal, 10.40 - 12.50 μm) (https://landsat.usgs.gov/).

The Landsat Enhanced Thematic Mapper Plus (**ETM**+) sensor is carried on Landsat 7, the

seventh satellite of the Landsat program. The observation bands are essentially the same seven bands as TM. The primary new features on Landsat 7 are a panchromatic band with 15 m spatial resolution, an on-board full aperture solar calibrator, 5% absolute radiometric calibration and a thermal IR channel with a four-fold improvement in spatial resolution over TM. An instrument malfunction occurred on May 31, 2003, with the result that all Landsat 7 scenes acquired since July 14, 2003 have been collected in "SLC-off" mode (https://landsat.usgs.gov/).

Landsat 8 carries two push-broom instruments: the Operational Land Imager (**OLI**), and the Thermal Infrared Sensor (TIRS). The spectral bands (Bands 2-8) of the **OLI** sensor, while similar to ETM+ sensor (Bands 1-5 and 7-8), provides enhancement from prior Landsat instruments, with the addition of two new spectral bands: a deep blue visible channel (Band 1 - Ultra Blue, 0.435 - 0.451 μm) specifically designed for water resources and coastal zone investigation, and a new infrared channel (Band 9 - Cirrus, 1.363 - 1.384 μm) for the detection of cirrus clouds. Thermal Band 10 (Thermal Infrared (TIRS 1, 10.60 - 11.19 μm)) and Band 10 (Thermal Infrared (TIRS 2, 11.50 - 12.51 μm)) are useful in providing more accurate surface temperatures and are collected at 100 meters (https://landsat.usgs.gov/).

The following figure shows the general view of detail of TM, ETM+ and OLI band designations.

[Figure]

The band designations for all Landsat sensors (https://landsat.usgs.gov/).

The data quality (signal to noise ratio) and radiometric quantization (12-bits) of the OLI and TIRS is higher than previous Landsat instruments (8-bit for TM and ETM+), providing significant improvement in the ability to detect changes on the Earth's surface. The principal functional

differences between the ETM+ and the former TM series are the addition of a 15 m resolution panchromatic band and two 8-bit "gain" ranges. The goal of using two gain settings is to maximize the sensors' 8-bit radiometric resolution without saturating the detectors while the L4 & L5 TM radiometric calibration uncertainty of the at-sensor spectral radiances due to change of gains was around 5% and was somewhat worse (Chander et al., 2009).

The details and difference of Landsat TM, ETM+ and OLI were already analyzed in the manuscript (Page 7, lines 21-30; Page 8, lines 1-8; Page 15, lines 3-11; Page 30, Table 4).

is more than 0.8 km and the farthest distance is about 23.98 km (Page 31, figure 1b), which covers a large area of Xuwen coast (10.77%), including two core areas and a buffer area in Xuwen Coral Reef Reserve. TSS concentrations of the 32 samples ranges from **4.3 mg/L** to **37.8 mg/L** with a mean value of **11.2 mg/L** in Xuwen coast. These samples represent the variability of TSS concentrations in the region well.

TSS concentrations in Moyangjiang River Estuary, Pearl River Estuary and Hanjiang River Estuary of Guangdong Province (Page 31, figure 1c, d, e) were mainly affected by the interaction of runoff and tide. So, the samples in the three estuaries (Page 31, figure 1c, d, e) with the **almost same representativeness** were carried out from the **lower river reaches** to the **outer shelf area** (including shoals, channels, maximum turbidity zones). TSS concentrations of the total of 62 simples (Page 31, figure 1c, d, e) in the three estuaries ranging from **5.2 mg/L** to **220.7 mg/L**. Among them, TSS concentrations of 15 simples ranges from 106 mg/L to 220.7 mg/L with a mean value of **150.9 mg/L**. In addition, the mean distance of two samples and the farthest distance from lower river reaches in the three estuaries are more than 0.7 km and about **15.3 km** in Moyangjiang River Estuary (Page 31, figure 1c), 1.3 km and **27.4 km** in Pearl River Estuary (Page 31, figure 1d) and 0.7 km and **8.7 km** in Hanjiang River Estuary (Page 31, figure 1e), respectively. We believe that the samples can reflect the characteristic of turbidity of estuaries of Guangdong Province well.

Besides, TSS concentrations of the 34 samples in Yangtze Estuary (Page 31, figure 1f) ranges from **38.4 mg/L** to **577.2 mg/L** with a mean value of 178.2 mg/L. The mean distance of two samples is more than **2.6 km** and the farthest distance from lower river reaches is about **73.32 km** (Page 31, figure 1f). These samples reflect the large variation of TSS concentrations of Yangtze River well.

Following your suggestions, we have added the statements to discuss and illustrate the representativeness of the location or concentrations of the samples (Page 7, lines 1-4).

*It is better to mention that root mean square of Trimble global positioning system is related to real time or post processed accuracy (page 6 line 21)*
**Response:**

The root mean square error of Trimble global positioning system is related to real time. We have revised the statement to make it clearer. (Page 7, line 1).

*The process of matching image and sample points should be discussed.*

**Response:**

Similar to previous studies (Bailey and Werdell. 2006; Chen et al., 2015; Zhang et al., 2014), the process of producing the satellite data for comparison with the in situ measurements was based on the positions of the samples recorded by Trimble global positioning system and considered the **two-hour** time window around the satellite overpass (Page 8, line 26; Page 8, lines 10-12, 21-22; Page 15, lines 15-18).

*Section 3.2, page 12 line 12. I am not clear how the vertex values have been calculated. This should be clarified.*

**Response:**

We are sorry for the omission of the calculation of the vertex values. We have added the following contents and revised the statements to make the calculation of the vertex values clearly. (Page 13, lines 16-21).

Firstly, we can get the derivative (equation (2)) from equation (1) by calculus. Then, we can obtain the vertex point through solving the roots of the derivative (equation (2)). Finally, the vertex values can be calculated by the parameters. Parameters a, b and c corresponding to OLI, ETM+ and TM sensors are -0.3575, 1.1135, 0.7162, -0.2844, 0.8578, 0.8278 and -0.2821, 0.8506, 0.8295, respectively (Page 34, figure 4). The unit of TSS concentration is in mg/L. So, the vertex values are 36.08633 mg/L (~36.1 mg/L) for OLI, 32.21716 mg/L (~32.2 mg/L) for ETM+ and 32.18162 mg/L (~32.2 mg/L) for TM (Page 13, lines 19-20).

$$\frac{Log(R_1)}{Log(R_2)} = a*(Log(TSS))^2 + b*Log(TSS) + c \quad (1)$$

$$\left(\frac{Log(R_1)}{Log(R_2)}\right)' = 2a*(Log(TSS)) + b$$

$$\Rightarrow Log(TSS) = -\frac{b}{2a} \qquad\qquad (2)$$

$$\Rightarrow TSS = 10^{-\frac{b}{2a}}$$

*Page 18, line 1. New results have been introduced in the summary and conclusions. These should be in the body of the text or in figures first.*

**Response:**

   Now we have revised some statements on the validation of QRLTSS model from Landsat imageries to illustrate these results (Section 3.4. Page 17, lines 17-20). In fact, the results are not new findings. These results had already been shown in figure 7d (Page 37).

*Technical corrections*
*There are numerous grammatical and structural issues that should be addressed. This manuscript would benefit from editing and review from a native English speaker to improve structure, flow and readability.*

**Response:**

   Thanks a lot for the valuable comments about the grammatical and structural issues. Following your suggestions, we have asked for **two native speakers** (associated professors of USA) helping us to pick up grammatical errors and revise the manuscript very carefully. **In addition**, reviewer #1 had also helped to point out main writing errors and structural issues and revised our manuscript carefully. In the revision process, the linkages had been polished and added in the manuscript for the improvement of the flow between sentences, and paragraphs, improving the understanding of the general modelling strategy (Page 3, line 19; Page 15, line 14; Page 6, lines 21-25; Page 9, lines 2-4; Section 2.1; Section 2.4; Section 2.5; Section 3.4). The contents of "extraction of water bodies" in Section 3.4 is removed to Section 2.1 (Page 6). The contents of the "Band response function application of Landsat sensors for field spectra" in Section 3.1 have been moved to Section 2.4 (Page 10). All the revising have been marked in red in the marked-up mode (Page 1, lines 1-2, 14-15, 17-23, 25-28; Page 2, lines 1-2, 5-6, 10; Page 3, lines 1-4, 15-16, 18-19, 26; Page 4, lines 20-22, 24-27; Page 5, lines 9-10, 17-18, 22-23; Page 6, lines 5-6, 13-14, 21-25; Page 7; Page 8, lines 1-6, 10-15, 17, 22, 26-27; Page 9, lines 1-3, 7-9, 12-15, 18-21; Page 10, lines

10-11, 14-16; Page 11, lines 6-7; Page 12, lines 2-4; Page 13, lines 14-21, 24; Page 14, lines 18-19; Page 15, lines 6-9, 19-22; Page 16, lines 4-9, 16, 26-27; Page 17, lines 6-7, 17-20, 23-24; Page 18, lines 4-5, 14, 19-23, 27; Page 19, lines 1-4, 6, 9-10, 17, 21-23).

*Section 2.2, I would recommend the dates being in day month year order to make them easier to read/follow.*
**Response:**

We have revised the statements of the dates in section 2.2 to make them easier to read and checked the manuscript following your suggestions. (Page 7, lines 4-8, 10, ; Page 8, lines 13-15, 17, 22; Page 15, lines 18-21; Page 16, line 4).

*Sentences should not start with the word 'And'*
*The writing is sometimes too conversational. For example, page 4 line 20 "What's more" could be replaced with "furthermore".*
**Response:**

Thanks for the comments. We have checked the manuscript carefully and revised the sentences that start with the word "And" (Page 4, lines 20-22, 26-27; Page 13, line 24; Page 15, lines 6-7; Page 16, lines 7-9).

Following your suggestions, we have revised the statements (Page 4, lines 20-22) that you pointed out and modified relevant writing of the whole manuscript carefully from native speakers.

*Tables 1-3 could be easier to read if each section was more clearly separated. For example, table 1, it is not clear what dates and samples are specific to Region c. Table 2, it is difficult to tell which turbidity models are used for TM 1,3, 4 as opposed to TM, 1,2,3,4. Table 3, I suggest adding a line between the previous models and the ones used in this study: : :etc*
*Figure 1 should go from a to e rather than e,a,b,c,d, as is currently the case. Need to make it clearer in the caption that the black dots are the sampling locations. Please explain the difference between circles and triangle in figure 1c (page 21).*
**Response:**

Following your suggestions, we added horizontal lines to the tables 1-3 for separating each section more clearly. Indeed, they are easier to read after the improvement.

We have revised the figure 1 (goes from a to e) and make it clearer that the **black dots and triangles** are all the sampling locations in the caption (Page 31). There are **40 samples** (dots and triangles) in figure 1c. Among them, the field **spectral measurements** and synchronous water samples (TSS concentrations) of **30 samples (dots)** were collected (Page 7, lines 8-9). Another **ten**

**samples (triangles)** in figure 1c have the **TSS concentrations collected only** (Page 7, lines 9-11). With or without synchronous spectral measurements is the difference between circles and triangles in figure 1c (Page 31).

The revised tables 1-3 and figure 1 has also been shown as follows.

Once again, thank you very much for your valuable comments and suggestions for the improvement of the manuscript!

**The revised Table 1.** Information about the study areas and in-situ data

| Location | Hydrologic features (length, drainage area, mean surface runoff and sediment discharge) | Date | Samples | Measurements | Number of synchronous samples with satellite |
|---|---|---|---|---|---|
| Region I | $-$, $-$ $8.6 \cdot 10^8$ m$^3$/year, $3 \cdot 10^4$ ton/year | Dec 3, 2010 | 10 | Reflectance, TSS | No |
| | | Jan 13~14, 2013 | 22 | Reflectance, TSS | No |
| Region II | 199 km, $6 \cdot 10^3$ km$^2$, $8.21 \cdot 10^9$ m$^3$/year, $3.27 \cdot 10^5$ ton/year | Dec 6, 2013 | 11 | Reflectance, TSS | 7, OLI |
| Region III | 2320 km, $4.53 \cdot 10^5$ km$^2$, $3.26 \cdot 10^{11}$ m$^3$/year, $7.53 \cdot 10^7$ ton/year | Dec 19, 2006 | 5 | Reflectance, TSS | No |
| | | Dec 21, 2006 | 18 | 8 samples with Reflectance and TSS; 10 samples with TSS only | 13, Hyperion |
| | | Dec 27, 2007 | 8 | Reflectance, TSS | No |
| | | Nov 2, 2012 | 9 | Reflectance, TSS | 6, ETM+ |
| Region IV | 470 km, $3.01 \cdot 10^4$ km$^2$, $2.45 \cdot 10^{10}$ m$^3$/year, $6.93 \cdot 10^6$ ton/year | Dec 1, 2013 | 12 | Reflectance, TSS | 9, OLI |
| Region V | 6280 km, $1.8 \cdot 10^6$ km$^2$, $9.2 \cdot 10^{11}$ m$^3$/year, $4.8 \cdot 10^8$ ton/year | Oct 14~15, 2009 | 34 | Reflectance, TSS | No |

**The revised Table 2.** Review of previous TSS or Turbidity retrieval models using Landsat imagery.

[revised manuscript text omitted]

| TM Band 1 | $Turbidity=19 \cdot B1 - 97.9$ | | |
| TM Band 2 | $Turbidity=6.4 \cdot B2 - 28$ | | |
| TM Band 3 | $TSS=69.39 \cdot B3 - 201$ | Ganges and Brahmaputra Rivers | Islam et al. (2001) |
| MSS Bands 1, 2 | $Ln(TSS)= 2.71 \cdot (B1/B2)^2 - 9.21 \cdot (B1/B2) + 8.45$ | Enid Reservoir in North Central Mississippi | Ritchie and Cooper. (1991) |
| TM Band 3 | $Log(TSS)= 0.098 \cdot B3 + 0.334$ | Delaware Bay | Keiner and Yan. (1998) |
| TM Bands 2, 3 | $TSS=0.7581 \cdot exp(61.683 \cdot (B2+B3)/2)$ | southern Frisian lakes | Dekkera et al. (2001) |
| TM Bands 1, 3 | $TSS=0.0167 \cdot exp(12.3 \cdot B3/B1)$ | an embayment of Lake Michigan | Lathrop et al. (1991) |
| TM/ETM+ Band 3 | $Log(TSS)= 44.072 \cdot B3 + 0.1591$ | Yellow River estuary | Zhang et al. (2014) |
| TM Band 3 | $TSS=2.19 \cdot exp(21.965 \cdot B3)$
 $TSS=-9275.78 \cdot (B3)^3 + 8623.19 \cdot (B3)^2$ | Poyang Lake | Wu et al. (2013) |

| | | | |
|---|---|---|---|
| | -810.04•B3+23.44 | | |
| TM Bands 3, 4 | TSS=5829.8•(B3-B4)$^3$+4165.09•(B3-B4)$^2$ -189.88•(B3-B4)+5.43 | | |
| | TSS=3.411•exp(21.998•(B3-B4)) | | |
| OLI Bands 2, 3, 8 | TSS=-191.02•B2+36.8•B3+172.66•B8+4.57 | Xin'anjiang Reservoir | Zhang et al. (2015) |
| TM Band 2 | B2=0.0044•TSS+2.5226 | Bhopal Upper Lake | Rao et al. (2009) |
| TM Band 2, 3 | Log(TSS)= 6.2244•(B2+B3)/B2•B3+0.892 | Yangtze estuary | Li et al. (2010) |
| TM Band 3 | TSS=0.543•B3-7.102 | Beysehir Lake | Nas et al. (2010) |
| TM Band 4 | TSS=229457.695•(B4)$^2$+146.462•B4+5.701 | Bohai gulf | Chen et al. (2014) |

**Table 3.** The comparison of calibration and validation accuracy of several best TSS retrieval models

| Model form | | Calibration ($R^2$) | Validation (RMSE(mg/L), MRE) | | |
|---|---|---|---|---|---|
| | | | Whole | Low range | High range |
| Chen et al. (2014) | | 0.7842 | 35.7, 144.2% | 18.35, 215.58% | 53.56, 23.5% |
| Li et al. (2010) | | 0.6167 | 69.3, 45% | 5.66, 52.6% | 113.48, 32.1% |
| Zhang et al. (2014) | | 0.5804 | 82.8, 48% | 6.56, 53.9% | 135.54, 38.15% |
| Lathrop et al. (1991) | | 0.6661 | 50.3 39.4% | 6.12, 44.6% | 82.09, 30.57% |
| Ritchie and Cooper. (1991) | | 0.6983 | 68.7, 41.3% | 7.24, 44% | 112.32, 36.6% |
| This study | OLI | 0.7181 | 21.5, 27.2% | 3.5, 31.1% | 35.1, 20.7% |
| | ETM+ | 0.708 | 25, 32.5% | 4.6, 38.3% | 40.7, 20.3% |
| | TM | 0.7079 | 24.9, 31.5% | 4, 35.3% | 40.6, 25.1% |

**Table 4.** The performance of different sensors and the vertexs of QRLTSS model based on these sensors

|  | TM | ETM+ | OLI | MODIS |
|---|---|---|---|---|
|  | Red band, Near infrared band | Red band, Near infrared band | Red band, Near infrared band | Red band, Near infrared band |
| Wavelength (nm) | 630-690, 760-900 | 630-690, 775-900 | 630-680, 845-885 | 620-670, 841-874 |
| Spatial resolution (m) | 30 | 30 | 30 | 250 |
| Radiometric resolution (bit) | 8 | 8 | 12 | 12 |
| Signal/noise (dB and Specified level of high ) | 140, 244 | 140, 244 | 340, 460 | 128, 201 |
| The vertex of quadratic model | 0.031, 32.2 mg/L (This study) | 0.031, 32.2 mg/L (This study) | 0.032, 36.1 mg/L (This study) | 0.025, 31 mg/L (Chen et al., 2015b) |

**Figures**

[Figure]

**Figure 1.** Study areas and locations of in situ data (black dots and triangles). b: Xuwen coast; c: Moyangjiang River Estuary;

d: Pearl River Estuary; e: Hanjiang River Estuary; f: Yangtze River Estuary.

[Figure]

**Figure 2.** 119 spectra were collected from study areas by ASD

[Figure]

**Figure 3.** The recalibration and validation of previous five TSS retrieval models based on 119 in situ samples. The models were developed by a (Chen et al., 2014), b (Li et al., 2010), c (Zhang et al., 2014), d (Lathrop et al., 1991), e (Ritchie and Cooper., 1991), respectively.

[Figure]

**Figure 4.** The calibration and validation results of TSS retrieval models: based on 119 in situ data (a) OLI, (b) ETM+ and (c) TM.

[Figure]

**Figure 5.** Relationship between the Landsat red band reflectance and corresponding TSS concentration. a: OLI sensor; b: ETM+ and TM sensors.

[Figure]

**Figure 6.** Scatterplot of Landsat measured reflectance versus in-situ reflectance. The former is calculated by averaging over a box of 3x3 pixels centered samples. a: red band; b: near infrared band.

[Figure]

**Figure 7.** Estimated TSS concentrations based on QRLTSS model in Moyangjiang River Estuary at 11:00 (Beijing time, OLI), on December 6, 2013 (a), Pearl River Estuary at 10:48 (Beijing time, ETM+), on November 2, 2012 (b), Hanjiang River Estuary at 10:41 (Beijing time, OLI), on December 1, 2013 (c), and comparison between the in-situ measured and Landsat imagery inversed TSS concentrations of three estuaries (d). Color scale is the legend of the TSS concentrations, in mg/L.

[Figure]

**Figure 8.** Estimated TSS concentrations based on QRLTSS model from EO-1 Hyperion imagery in Pearl River Estuary at 10:33 (Beijing time), on December 21, 2006 (a) and comparison between the in-situ measured and EO-1 Hyperion imagery inversed TSS concentrations (b). Color scale is the legend of the TSS concentrations, in mg/L.